

# Global agricultural ammonia emissions simulated with the ORCHIDEE land surface model

Maureen Beaudor[1], Nicolas Vuichard[1], Juliette Lathière[1], Nikolaos Evangeliou[2], Martin Van Damme[3,4], Lieven Clarisse[3], and Didier Hauglustaine[1]

[1]Laboratoire des Sciences du Climat et de l'Environnement (LSCE) CEA-CNRS-UVSQ, Gif-sur-Yvette, France
[2]Norwegian Institute for Air Research (NILU), Department of Atmospheric and Climate Research (ATMOS), Kjeller, Norway.
[3]Université libre de Bruxelles (ULB), Spectroscopy, Quantum Chemistry and Atmospheric Remote Sensing (SQUARES), Brussels, Belgium
[4]Royal Belgian Institute for Space Aeronomy, Brussels, Belgium

**Correspondence:** Maureen Beaudor (maureen.beaudor@lsce.ipsl.fr)

**Abstract.**

Ammonia ($NH_3$) is an important atmospheric constituent. It plays a role in air quality and climate through the formation of ammonium sulfate and ammonium nitrate particles. It has also an impact on ecosystems through deposition processes. About 85% of $NH_3$ global anthropogenic emissions are related to food and feed production and, in particular, to the use of mineral fertilizers and manure management. Most global chemistry transport models rely on bottom-up emission inventories subject to significant uncertainties. In this study, we estimate emissions from livestock by developing a new module to calculate ammonia emissions coming from the whole agricultural sector (from housing and storage to grazing and fertilizer applications) within the global land surface model ORCHIDEE. We detail the approach used for quantifying livestock feeding management, manure applications and indoor and soil emissions and evaluate the model performance. Our results reflect China, India, Africa, Latin America, the USA, and Europe as the main contributors to the global $NH_3$ emissions accounting for 80 % of the total budget. The global calculated emissions reach $44 \mathrm{TgN yr}^{-1}$ over the 2005-2015 period, which is within the range estimated by previous work. Key parameters (pH of the manure, timing of the N application, atmospheric $NH_3$ surface concentration, etc . . . ) which drive the soil emissions have also been tested in order to assess the sensibility of our model. Manure pH is the parameter to which modeled emissions are the most sensitive to with a 10% change in emissions per % change in pH. Even though we found an under-estimation in our emissions over Europe (-26%) and an over-estimation in the USA (+56%) compared to previous work, other hot-spot regions are consistent. The calculated emissions seasonality is in very good agreement with satellite based emissions. These encouraging results prove the potential of coupling ORCHIDEE land-based emissions to CTMs, which are currently forced by bottom-up anthropogenic-centered inventories such as CEDS.

## 1 Introduction

Ammonia ($NH_3$) is a crucial species in the atmosphere playing a role in the alteration of air quality and climate through its implication in airborne particle matter formation (PM or aerosols) (Anderson et al., 2003; Bauer et al., 2007). $NH_3$ lifetime is





short and has been reported to range from a few hours to a few days (Pinder et al., 2008; Behera et al., 2013) since ammonia mostly originates from surface emissions and its deposition velocity is high over most surfaces (Hov et al., 1994; Evangeliou et al., 2020). Due to this characteristic, $NH_3$ is transported over relatively short distances and readily reacts with abundant gases

such as nitric and sulfuric acids to form secondary aerosols (Malm et al., 2004). The resulting aerosols, such as ammonium nitrates or ammonium sulfates, have important impacts on the Earth's radiative budget due to their ability to scatter the incoming radiation, act as cloud condensation nuclei, and indirectly increase cloud lifetime (Abbatt et al., 2006; Henze et al., 2012; Behera et al., 2013; Evangeliou et al., 2020). The impact of $NH_3$ on the total radiative forcing is estimated at $-0.06 W.m^{-2}$ by contributing to the radiative forcing of the nitrate and the sulfate of about $-0.07$ and $0.01 W.m^{-2}$ respectively (Myhre

et al., 2013). By analyzing different representative concentration pathways (RCP) scenarios, Hauglustaine et al. (2014) have shown the importance of ammonia on the direct aerosol forcing in the future due to the potentially significant increase in the agricultural emissions. In the most extreme scenario, emissions can increase by 50% by 2100 compared to their present-day level.

In addition to its contribution to the radiative budget, the balance between $NH_3$, $SO_2$, $NO_x$ emissions controls the formation

of secondary inorganic aerosol (SIA), important components of fine particles ($PM_{2.5}$) (Paulot et al., 2016; Fu et al., 2017; Sutton et al., 2020). Quantifying ammonia emissions is of high interest for air quality policies since it appears that $NH_3$ emission reductions would also be efficient to reduce inorganic aerosol formation (Lachatre et al., 2019).

There are many issues in the development of reliable $NH_3$ emission inventories, as analyzed by Nair and Yu (2020), such as the lack of emission measurements, the difficulties in validating $NH_3$ concentration with measurements, the critical as-

sumptions behind the modeling approaches in terms of emission factors and activity rates. Even though ammonia emissions are challenging to estimate, several studies aimed at quantifying global emissions and their associated uncertainty. For example, Dentener and Crutzen (1994) estimate a global $NH_3$ emission of $45 TgNyr^{-1}$ (+/-50%), a low estimate compared to the $54 TgNyr^{-1}$ (+/-25%) of Bouwman et al. (2005) and the $75 TgNyr^{-1}$ (+/- 50%) of Schlesinger and Hartley (1992) (Zhu et al., 2015). Agricultural activities are among the significant sources of ammonia in the world, accounting for about 85% of

the global anthropogenic $NH_3$ emissions (Behera et al., 2013). Agricultural emissions originate from fertilizer application and livestock management, the latter including livestock housing, manure storage, and manure applications. Globally, recent studies developed methodologies in order to quantify emissions from this sector. For example, Beusen et al. (2008), Paulot et al. (2014) and Hoesly et al. (2018) estimated similar emissions of about $32-35 TgNyr^{-1}$, which is less than the $41-47 TgNyr^{-1}$ estimates of Crippa et al. (2018) and Vira et al. (2019).

Modeling $NH_3$ sources from agriculture is especially difficult since it depends on several factors related to the environment (atmospheric conditions, soil properties), and to agricultural practices, which are also crucial to capture the temporal and spatial variability of emissions correctly. Emissions from manure management are driven by the amount of nitrogen in the feeding, animal body characteristics, housing conditions of the animal, temperature, and animal waste handling practices (Anderson et al., 2003). The soil $NH_3$ emissions originate from N application either from fertilizer or manure and are controlled by the

soil pH, temperature, water content, surface wind speed and atmospheric $NH_3$ concentration (Kirk and Nye, 1991; Cellier





et al., 2011; Behera et al., 2013). Other factors such as the ammonium content of the fertilizer and the timing of N application are also crucial for emission estimates (Riddick et al., 2016; Vira et al., 2019).

A first type of approach in the quantification of agricultural ammonia emissions is the bottom-up inventories. Most global inventories, such as CEDS (Hoesly et al., 2018), EDGAR (Crippa et al., 2018) and HTAP (Janssens-Maenhout et al., 2015) are based on activity data associated with a corresponding emission factors (EF). Chemistry Transport Models (CTMs) are usually forced with these global emission inventories. As examples, the inventory described by Bouwman et al. (1997) is prescribed in the study of Xu and Penner (2012) and the Community Emissions Data System (CEDS) inventory (Hoesly et al., 2018) is used in Paulot et al. (2016) and Pai et al. (2021). Emission inventories do not account for environmental factors such as the temperature or the soil humidity, which is an important limitation for studying spatial-temporal variability of atmospheric $NH_3$ and $NH_4^+$ concentrations. Most inventories rely on fertilizer application period to represent the seasonality of emissions but are based on few studies and usually use the same temporal profile (most of the time reflecting European agricultural practices), which is extrapolated to the whole globe. More complex inventories exist, such as the updated version of the Global Livestock Environmental Assessment Model (GLEAM) (Uwizeye et al., 2020) or the comprehensive food system developed by Conijn et al. (2018) and combine more detailed agricultural information (animal requirements, livestock system types, manure management handlings, surface types receiving manure) with EF but consider yearly emissions. Even though this type of approach is more accurate due to the detailed consideration of agricultural practices, it shows limitations for studying the temporal variability of emissions due to the static representation of the agricultural practices when using unique EF or only one seasonal profile for the whole globe. Recently, more complex models based on an explicit description of processes that control the volatilization from soil have been developed. The FAN model initially developed by Riddick et al. (2016) and largely improved by Vira et al. (2019) combines information on agricultural practices, emission factors for manure management emissions, and physical processes for soil volatilization to compute $NH_3$ emissions from the different agricultural sources. When soil processes are tightly coupled to the main meteorological drivers, the related emissions respond to environmental changes, which is particularly interesting in the case of climate-surface interaction studies. Even if the FAN model is integrated into the Community Earth System Model (CESM), the manure produced by the livestock is not directly linked to the biomass productivity, which can represent uncertainty in the nitrogen content of the manure and, therefore, in the resulting emissions.

In this study, in order to better account for the key parameters in the estimate of the $NH_3$ emissions, we implement a module representing the agricultural sector within the Land Surface Model (LSM) ORCHIDEE. Our methodology is based on the integration of a complete dynamical agricultural module (CAMEO) within ORCHIDEE, which details a feeding management module linked to the biomass productivity of the model and animal characteristic information, a manure management representation that combines regional agricultural handling practices and a complex soil emission component based on key environmental parameters such as the vegetation growth, temperature, and soil humidity.

Section 2 describes the agricultural model within ORCHIDEE, and the model set up of the 11 years control simulation (2005-2015), along with the sensitivity analysis simulation set. Global and regional results by comparing previous works (CEDS and the FAN model from (Vira et al., 2019)) and seasonal analysis using airborne measurements (IASI derived emissions) are presented and discussed in Section 3. The conclusions are provided in Section 4.



## 2    Methods

This section describes the process-based model for the nitrogen flow coming from agriculture within the LSM ORCHIDEE. The new module implemented aims at calculating two types of emissions from agriculture : the manure management chain emissions (livestock housing and yard, and manure storage) and the soil emissions (accounting for the fertilizer and manure applications). The ORCHIDEE model framework is described in Section 2.1.1 followed by the different interactive components (shown in Fig 1) : the feeding of livestock (Section 2.1.2.1), the whole manure management chain (Section 2.1.2.2), the fertilizer surface application (Section 2.1.2.3) and the soil-plant-atmosphere continuum processes leading to soil emissions (Section 2.1.2.4). Section 2.2 describes the set-up of the simulations and the model evaluation protocol.

### 2.1    The ORCHIDEE Land Surface Model

#### 2.1.1    The General description

ORCHIDEE is a global-scale terrestrial ecosystem model coupling energy, water and both carbon and nitrogen cycles (Ciais et al., 2005; Krinner et al., 2005; Piao et al., 2007). The vegetation is represented by 15 Plant Functional Types (PFTs), among which two crop types (C3 and C4) and four grass types (temperate, boreal, and tropical C3 grasses and a single C4 class). The initial version used in this study includes a simple management of the crop biomass (which assumes that 45% of the Net Primary Productivity (NPP) is harvested) but no grassland management.

The main nitrogen processes within the soil-plant-atmosphere continuum are based on the OCN model (Zaehle and Friend, 2010; Zaehle et al., 2010). The representation of nitrification and denitrification processes are based on the DNDC model (Li et al., 1992; Li, 2000; Zhang et al., 2002). It accounts for ammonia/ammonium ($NH_3$ / $NH_4^+$), nitrate ($NO_3^-$), nitrogen oxides ($NO_x$) and nitrous oxide ($N_2O$) pools and the related emissions. In addition to $NH_3$, $NO_x$, $N_2O$ and $N_2$ emissions, N is lost through run-off and leaching processes. The N inputs to soil mineral pools include the atmospheric $NO_y$ and $NH_x$ depositions, the Biological Nitrogen Fixation (BNF), and the application of synthetic and organic fertilizers over agricultural lands. The version of ORCHIDEE used for this study is ORCHIDEEv3 revision 6863. It was part of the ensemble of Terrestrial Ecosystem Models used for the 2019 Global Carbon Budget (Friedlingstein et al., 2019) and was recently evaluated by Seiler et al. (in prep). Overall, ORCHIDEEv3 shows a good agreement with observation-based data for carbon fluxes and vegetation state. Former revisions of ORCHIDEEv3 have also been used to quantify globally the GPP flux (Vuichard et al., 2019) and soil $N_2O$ emissions (Tian et al., 2018). In the initial version, the organic fertilizer (i.e., manure) amount was prescribed annually (Zhang et al., 2017a) and the corresponding quantity of N was applied at a constant rate daily over the whole year. In addition, the emissions from the whole manure management were missing, and only soil emissions were taken into account. A description of the ORCHIDEE model, including the nitrogen cycle and its interaction with the carbon cycle, is detailed in Vuichard et al. (2019). The model evaluation shows at global scale a good agreement between the gross primary production simulated with the carbon-nitrogen interaction version and the observational validation set.





In this paper, we integrate the following new developments within a new module called CAMEO for Calculation of AMmonia Emissions in ORCHIDEE which are detailed hereafter :

- a new grassland and cropland management module dedicated to livestock feeding (Section 2.1.2.1);

125
- a module computing the manure production and the associated emissions from the indoor farming livestock activities (housing, yard, manure storage) (Section 2.1.2.2).

- a new parametrization for the agricultural N application onto croplands and grasslands (Section 2.1.2.3).

- an improved soil emission scheme based on more realistic representation of the soil-plant-atmosphere continuum (Section 2.1.2.4).

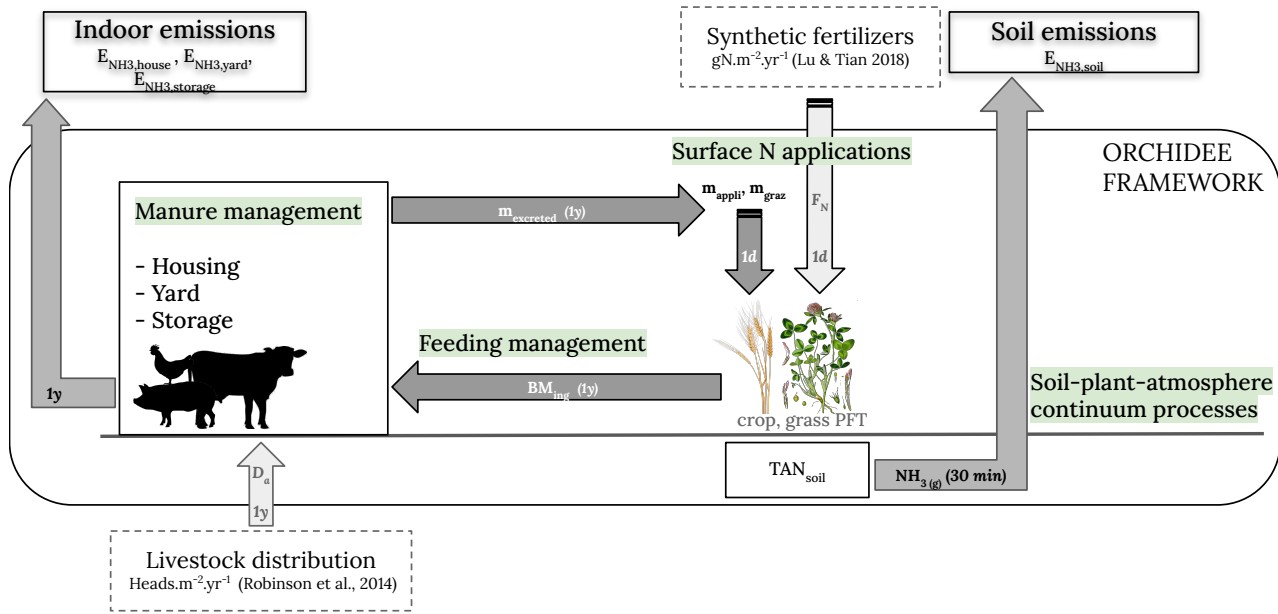

**Figure 1.** Scheme of the agro-ecosystem representation developed within ORCHIDEE C-N. It is composed of four main components describing the feeding management, the manure management, the surface N applications and the soil-plant-atmosphere continuum processes leading to the soil emissions. The livestock distributions and the synthetic fertilizers are the main forcing files of the system and are represented in the dashed frames. The time-steps (1y, 1d and 30 min) of the processes are indicated in the arrows.

.



### 2.1.2 Agricultural N-flow module within ORCHIDEE : CAMEO

#### 2.1.2.1 Livestock feeding management

Both the feeding ($BM_{ing}$, $kgCm^{-2}yr^{-1}$) and bedding ($BM_{bedding}$, $kgCm^{-2}yr^{-1}$) needs are calculated within each grid cell from livestock density distribution maps, for different livestock categories. The livestock types considered in our study are the non-dairy cattle, the dairy cattle, the pig, the small ruminants, and the chicken, which are the main contributors to the global livestock $NH_3$ emissions. $D_a$, the distribution of each livestock category 'a' is taken from the Gridded Livestock of the World (GLW 2 (Robinson et al., 2014)) for the year 2006.

$BM_{ing}$ for livestock category 'a' is calculated as followed :

$$BM_{ing\,(a)} = D_a \times SI \times W_a \tag{1}$$

where $W_a$ is the animal weight (kgC) and SI is the specific intake (the intake per animal weight unit, $kgCkg^{-1}yr^{-1}$).

A daily dry matter intake equal to 2.5% of the livestock weight (Paustian et al., 2006), is considered for every livestock categories, and a factor of 0.45 is used to convert the dry matter into carbon matter (Paustian et al., 2006), leading to a SI value of 0.01 $kgCkg^{-1}yr^{-1}$. Regarding livestock weights, we use regional values adapted from the supplement of FAO (2018), as listed in Table 1.

**Table 1.** Regional weights of the animal (kg) used in the intake needs calculation. Data have been adapted from FAO (2018). Regions: NA (North America), RUS (Russian Federation), WE (Western Europe), EE (Eastern Europe), NENA (Near East and North Africa), ESEA (East and Southeast Asia), OCE (Oceania), SA (South Asia), LAC (Latin America and the Caribbean) and SSA (Sub-Saharan Africa).

|                  | NA  | RUS | WE  | EE  | NENA | ESEA | OCE | SA  | LAC | SSA |
|------------------|-----|-----|-----|-----|------|------|-----|-----|-----|-----|
| Dairy cattle     | 750 | 500 | 594 | 514 | 370  | 398  | 461 | 336 | 556 | 287 |
| Non dairy cattle | 744 | 611 | 611 | 610 | 407  | 482  | 440 | 409 | 556 | 296 |
| Pig              | 157 | 142 | 163 | 148 | 117  | 103  | 113 | 91  | 143 | 72  |
| Chicken          | 1,5 | 1,7 | 1,9 | 1,8 | 1,6  | 1,7  | 1,7 | 1,4 | 1,6 | 1,7 |
| Small ruminant   | 85  | 77  | 75  | 76  | 50   | 40   | 76  | 39  | 60  | 34  |

The livestock feeding and bedding needs are provided by a fraction of crop and grass NPP which is harvested.

In order to quantify the amount of grassland and cropland biomass needed to feed each livestock category ($BM_{ing,grass\,(a)}$ and $BM_{ing,crop\,(a)}$, respectively), we use the fractions of grass and crop which constitute the diet composition of each animal ($d_{grass\,(a)}$ and $d_{crop\,(a)}$). The ruminant animals (i.e., cattle and small ruminants) have a diet composed of a portion of grass and



crop, the non-ruminant animals (i.e., pig and chicken) have a crop-only diet. The bedding needs are taken from crop residues
only.

$$\mathrm{BM_{ing,grass(a)} = BM_{ing(a)} \times d_{grass\,(a)}}$$

$$\mathrm{BM_{ing,crop(a)} = BM_{ing(a)} \times d_{crop\,(a)}} \qquad (2)$$

The diet composition $d_{\mathrm{crop/grass}\,(a)}$ is calculated from regional feeding information detailed in the Global Livestock Environmental Assessment Model (FAO, 2018) and described in the Supplementary Material. The bedding is estimated with EMEP/EEA (2019) :

$$\mathrm{BM_{bedding\,(a)} = D_a \times 0.32 \times Straw_a} \qquad (3)$$

$\mathrm{Straw}_a$ corresponds to the amount of straw ($\mathrm{kgHead^{-1}yr^{-1}}$) used as bedding for each livestock type (Table 2). The 0.32 factor corresponds to the C content of straw assuming a C:N ratio of 80 (USDA) for the straw material and a nitrogen content of $4\mathrm{gNkg^{-1}}$ (EMEP/EEA, 2019). This value is consistent with recent experimental studies by Su et al. (2020) where they found $0.35\mathrm{kgkg^{-1}}$ of total carbon in wheat straw.

$\mathrm{BM_{bedding\,(a)}}$ and $\mathrm{BM_{ing,crop(a)}}$ constitute the total demand in crop. We assume that the demand in crop biomass in each grid cell is satisfied by the amount of crop biomass harvested globally (global market). In contrast, the grass biomass needs are satisfied locally. Indeed, the grass biomass needs define the grassland management intensity through a grazing indicator (GI, unitless) which corresponds to the fraction of grass NPP for the year $y$ that is harvested. GI is defined as :

$$\mathrm{GI}_{(y)} = min(\frac{\mathrm{BM_{ing\,grass}}}{\mathrm{NPP_{grass\,above(y)}}}; max_{above}) \qquad (4)$$


where $\mathrm{NPP_{grass\,above(y)}}$ corresponds to the above NPP of grasslands at the grid cell level ($\mathrm{kgCm^{-2}[gridcell]yr^{-1}}$) and $max_{above}$, a parameter equal to 0.7 and defined as the maximum of the above biomass available for grazing/cutting.

$\mathrm{NPP_{grass\,above(y)}}$ is a function of the grassland NPP ($\mathrm{kgCm^{-2}[grassland]yr^{-1}}$) but also of the grassland area defined in
each grid-cell. Due to an inconsistency between the land-use map and livestock density map, the targeted $\mathrm{BM_{ing\,grass}}$ value may not be reached by the use of GI. To ensure that $\mathrm{BM_{ing\,grass}}$ demand is always satisfied, we adjust the diet composition of ruminants in some grid cells by increasing as much as needed $d_{\mathrm{crop}\,(a)}$ (and by reducing by the same factor $d_{\mathrm{grass}\,(a)}$). The adjusted value of $d_{\mathrm{grass}\,(a)}$ is named $d_{\mathrm{grass,\,adjusted}\,(a)}$ and depicted in the Supplementary Material (Fig S2.). GI is then applied to $\mathrm{NPP_{grass\,above(y)}}$ on a daily basis in order to obtain the total effective grazed biomass. $d_{\mathrm{grass,\,adjusted}\,(a)}$ is used to deduce
for each animal the effective crop biomass from the effective grazed biomass. Finally, each grid-cell's effective crop biomass is constrained by the global crop harvested NPP.

Since our methodology is based on a nitrogen flow scheme, the C:N ratio imposed by the model for the crop and grass products is used to convert the carbon into nitrogen biomass ingested (unit:$\mathrm{kgNm^{-2}yr^{-1}}$). Grassland C:N ratio is unique



for each grid-cell and varies spatially from 23 to 62, while cropland C:N is fixed for the whole globe and estimated to ~38.

$\mathrm{BM_{ing\,tot(y,a)}}$ represents the total (including crop and grass products) nitrogen biomass ingested, which is used to compute the resulting manure emissions (described in the next section). Concerning the crop used as straw, a fixed C:N ratio of 80 is chosen (EMEP/EEA, 2019).

### 2.1.2.2    Indoor N flows and ammonia emissions

We adapt the scheme developed by Dämmgen and Hutchings (2008) which defines indoor ammonia emissions for each animal category. These pathways have also been used in the Tier 2 methodology of the manure management part of the EMEP/EEA Air Pollutant Emission Inventory Guidebook EMEP/EEA (2019). It is based on an N-flow model with mass transfers and emissions proportional to the Total Ammonia Nitrogen (TAN).

The main output of this module is the N emissions that occur during housing, yard, and storage of the manure, along with
the resulted manure produced. The seasonal variability of indoor N emissions is neglected, and the emissions and the manure flow are calculated yearly.

Firstly, we compute the nitrogen biomass excreted by each animal category ($m_{\mathrm{excreted}}$) based on the excretion rate estimated by Paustian et al. (2006) for the IPCC Tier 2 recommendations (see Table 2).

$$m_{excreted(a)} = \mathrm{BM_{ing\,tot(y,a)}} \times N_{\mathrm{excretion\,rate(a)}} \tag{5}$$

Secondly, we compute the manure excreted during the different livestock activities as a proportion of the year spent in housing, yard, and grazing, based on EMEP/EEA (2019). The fraction of time spent at yard ($x_{\mathrm{yard}}$, Table 2) is prescribed. The remaining time fraction is split into grazing ($x_{\mathrm{graz}}$, Table 2) and housing periods.

$$m_{\mathrm{N\,(yard,a)}} = x_{\mathrm{yard,a}} \times m_{\mathrm{excreted(a)}}$$
$$m_{\mathrm{N\,(graz,a)}} = x_{\mathrm{graz,a}} \times (1 - x_{\mathrm{yard,a}}) \times m_{\mathrm{excreted(a)}}$$
$$m_{\mathrm{N\,(house,a)}} = (1 - x_{\mathrm{graz,a}}) \times (1 - x_{\mathrm{yard,a}}) \times m_{\mathrm{excreted(a)}} \tag{6}$$

Default values of the TAN fraction contained in the excretal N ($x_{\mathrm{TAN,a}}$) from the manure management part of the EMEP/EEA
Air Pollutant Emission Inventory Guidebook 2019 (EMEP/EEA, 2019) (see Table 2) are used to calculate the amount of TAN produced during each activity, i (housing, yard, and grazing):

$$m_{\mathrm{TAN\,(i,a)}} = x_{\mathrm{TAN,a}} \times m_{\mathrm{N(i,a)}} \tag{7}$$

$m_{\mathrm{TAN\,(graz,a)}}$ and $m_{\mathrm{N\,(graz,a)}}$ are used in Section 2.1.2.3 for N application on cultivated areas.

The ammonia emissions in house $E_{\mathrm{NH_3(house,a)}}$ (unit : $\mathrm{kgNm^{-2}yr^{-1}}$) combine the volatilizations from liquid and solid TAN
masses, with specific emission factors $\mathrm{EF_{NH_3\,(house,liq,a)}}$ ($\mathrm{NH_3 - N\,kgTAN^{-1}}$) and $\mathrm{EF_{NH_3\,(house,sol,a)}}$ ($\mathrm{NH_3 - N\,kgTAN^{-1}}$).



**Table 2.** Default values for the fractions of the year spent at grazing and yard, the proportion of TAN in the N mass excreted and the straw used in bedding based. The information has been taken and adapted from EMEP/EEA (2019). The straw used in bedding based for swine are the average between different livestock types. The N retained is taken from Paustian et al. (2006).

| | $x_{\text{graz}}$ (-) | $x_{\text{yard}}$ (-) | N retained (-) | $x_{\text{TAN}}$ (%) | Straw (kgHead$^{-1}$yr$^{-1}$) |
|---|---|---|---|---|---|
| Dairy cattle | 0.5 | 0.25 | 0.20 | 60 | 1500 |
| Non dairy cattle | 0.5 | 0.10 | 0.07 | 60 | 500 |
| Pig | 0 | 0 | 0.30 | 70 | 400 |
| Chicken | 0 | 0 | 0.30 | 70 | 0.00 |
| Small ruminants | 0.92 | 0.02 | 0.10 | 50 | 20 |

**Table 3.** Emission factors (EF) given as $\text{NH}_3 - \text{N kgTAN}^{-1}$. EF for the yard and the other N species come from EMEP/EEA (2019). The other EF are taken from Sommer et al. (2019). There is no distinction between liquid and solid manure for yard EF. Numbers in parenthesis are the standard deviation given in Sommer et al. (2019) and used for the sensitivity analysis.

| | Manure type | $\text{EF}_{\text{NH}_3 \text{ (house)}}$ | $\text{EF}_{\text{NH}_3 \text{ (yard)}}$ | $\text{EF}_{\text{NH}_3 \text{ (store)}}$ | $\text{EF}_{\text{N}_2 \text{ (store)}}$ | $\text{EF}_{\text{NO (store)}}$ | $\text{EF}_{\text{N}_2\text{O (store)}}$ |
|---|---|---|---|---|---|---|---|
| Dairy cattle | liquid | 19 (5.7) | 30 | 25 (11.2) | 0,3 | 0,01 | 0 |
| Dairy cattle | solid | 8 (5.7) | 30 | 32 (15.8) | 30 | 1 | 2 |
| Non dairy cattle | liquid | 19 (5.7) | 53 | 25 (11.2) | 0,3 | 0,01 | 0 |
| Non dairy cattle | solid | 8 (5.7) | 53 | 32 (15.8) | 30 | 1 | 2 |
| Swine | liquid | 27 (12.1) | 0 | 11 (6.9) | 0,3 | 0,01 | 0 |
| Pig | solid | 23 (12.6) | 0 | 29 (15.6) | 30 | 1 | 1 |
| Chicken | solid | 21 (11.5) | 0 | 19 (15.9) | 30 | 1 | 0,2 |
| Small ruminants | solid | 22 (5.7) | 75 | 30 (15.8) | 30 | 1 | 2 |

$\text{EF}_{\text{NH}_3 \text{ (house,liq,a)}}$ and $\text{EF}_{\text{NH}_3 \text{ (house,sol,a)}}$ values are taken from Sommer et al. (2019) for each animal a except for small ruminants which comes from EMEP/EEA (2019) (see Table 3). $E_{\text{NH}_3\text{(house,a)}}$ is written as:

$$E_{\text{NH}_3\text{(house,a)}} = (x_{\text{liq,a}} \times \text{EF}_{\text{NH}_3 \text{ (house,liq,a)}} + (1 - x_{\text{liq,a}}) \times \text{EF}_{\text{NH}_3 \text{ (house,sol,a)}}) \times m_{\text{TAN (house,a)}} \tag{8}$$

with $x_{\text{liq,a}}$ (unitless) the proportion of manure handled as liquid for livestock type 'a', adapted from the Global Livestock Environmental Assessment Model (FAO, 2018) (see Supplementary Material). **??** Emissions from yard ($E_{\text{NH}_3\text{(yard,a)}}$) are calculated from the mass excreted in yard and there is no distinction between liquid, and solid handling.

$$E_{\text{NH}_3\text{(yard,a)}} = \text{EF}_{\text{NH}_3 \text{ (yard,a)}} \times m_{\text{TAN (yard,a)}} \tag{9}$$

We compute the amounts of N and TAN that are stored as liquid and solid before application ($m_{\text{N (stor,type,a)}}$ and $m_{\text{TAN (stor,type,a)}}$ for type=liq,sol, respectively, $\text{kgNm}^{-2}\text{yr}^{-1}$, eq. 10). For storage, we assume that all the manure from house and yard is stored, except the nitrogen lost by ammonia emissions in house and yard ($E_{\text{NH}_3\text{(house,a)}}$ and $E_{\text{NH}_3\text{(yard,a)}}$). Manure from yard is considered liquid and goes in the liquid manure storage. Concerning the liquid storage (ie. slurries), a fraction $f_{\text{min}}$ of the organic N (N-TAN) is converted into TAN through mineralization. A value of 0.1 is used for $f_{\text{min}}$ (Dämmgen and Hutchings,





2008; EMEP/EEA, 2019). For solid storage, we account for an additional N source from bedding ($m_{\text{bed},N,a}$). Incorporation of bedding in the manure storage induces an immobilization of TAN in the organic matter when manure is handled as straw-based

solid manure, at a rate $f_{\text{imm}}$ proportional to $m_{\text{bed},N,a}$. A $f_{\text{imm}}$ value of $0.0067\ \text{kgkg}^{-1}$ is used (Kirchmann and Witter, 1989; Webb and Misselbrook, 2004; EMEP/EEA, 2019). This immobilization highly reduces the resulting $NH_3$ emissions.

$$m_{\text{N (stor,liq,a)}} = (m_{\text{N (house,liq,a)}} - E_{\text{NH}_3\text{(house,liq,a)}}) + (m_{\text{N (yard,a)}} - E_{\text{NH}_3\text{(yard,a)}})$$

$$m_{\text{TAN (stor,liq,a)}} = (m_{\text{TAN (house,liq,a)}} - E_{\text{NH}_3\text{(house,liq,a)}}) \times (1 - f_{min}) +$$

$$m_{\text{TAN (stor,liq,a)}} \times f_{min} + (m_{\text{TAN (yard,a)}} - E_{\text{NH}_3\text{(yard,a)}})$$

$$m_{\text{N (stor,sol,a)}} = m_{\text{N (house,sol,a)}} - E_{\text{NH}_3\text{(house,sol,a)}} + m_{\text{bed,N,a}}$$

$$m_{\text{TAN (stor,sol,a)}} = m_{\text{TAN (house,sol,a)}} - E_{\text{NH}_3\text{(house,sol,a)}} - m_{\text{bed,a}} \times f_{\text{imm}} \tag{10}$$

with $m_{\text{bed,N}}$, the N mass of bedding ($\text{kgNm}^{-2}\text{yr}^{-1}$) and $m_{\text{bed}}$ the dry matter mass of bedding.

Manure coming from storage is supposed to be entirely used as fertilizer. The quantities $m_{\text{TAN (applic)}}$ and $m_{\text{N (applic)}}$ are

the TAN and N manures which will be applied to surface as describe in Section 2.1.2.3. They are obtained by removing the total N emissions from the stored manure (Eq 11):

$$\textbf{liquid}\begin{cases} m_{\text{TAN (applic,liq,a)}} = m_{\text{TAN (stor,liq,a)}} - E_{\text{stor(liq,a)}} \\ m_{\text{N (applic,liq,a)}} = m_{\text{N (stor,liq,a)}} - E_{\text{stor(liq,a)}} \end{cases}$$

$$\textbf{solid}\begin{cases} m_{\text{TAN (applic,sol,a)}} = m_{\text{TAN(stor,sol,a)}} - E_{\text{stor(sol,a)}} \\ m_{\text{N (applic,sol,a)}} = m_{\text{N(stor,sol,,a)}} - E_{\text{stor(sol,a)}} \end{cases} \tag{11}$$

In addition to emissions of $NH_3$, other N species ($N_2O$, $NO$ and $N_2$) emissions can occur from storage and thus are required to calculate the final manure mass from storage. These emissions are obtained using the EFs listed in Table 3 as :

$$E_{\text{stor,liq,a}} = m_{\text{TAN (stor,liq,a)}} \times$$

$$(\text{EF}_{\text{NH}_3\text{ (stor,liq,a)}} + \text{EF}_{\text{N}_2\text{O (store,liq,a)}} + \text{EF}_{\text{NO (stor,liq,a)}} + \text{EF}_{\text{N}_2\text{ (stor,liq,a)}})$$

$$E_{\text{stor,sol,a}} = m_{\text{TAN (stor,sol,a)}} \times$$

$$(\text{EF}_{\text{NH}_3\text{ (stor,sol,a)}} + \text{EF}_{\text{N}_2\text{O (stor,sol,a)}} + \text{EF}_{\text{NO (stor,sol,a)}} + \text{EF}_{\text{N}_2\text{ (stor,sol,a)}})$$

$\tag{12}$

The remaining manure after storage $m_{\text{(applic,a)}}$ and the one produced during grazing $m_{\text{(graz,a)}}$ are the main output of this specific module. Both quantities are the input for the surface application component of the model (described in the next section).





**Table 4.** Summary of data sources used in sections 2.1.2.1 and 2.1.2.2 for the calculation of the indoor emissions. All the data is used for each livestock type a except for the variable $D_{\text{dairy cattle,i}}$.

| Abbreviation | Description | Unit | Sources |
|---|---|---|---|
| D | Spatial distribution for 2006 | Head/km$^2$ | Robinson et al. (2014) |
| $D_{\text{dairy cattle,i}}$ | Country level $i$ annual dairy cattle stocks | Head | FAOSTAT (2020) |
| W | Regional typical animal weight | kg | adapted from FAO (2018) |
| $d_{\text{crop/grass}}$ | Regional diet composition | % | adapted from FAO (2018) |
| Straw | Annual straw used in bedding | kg FM Head$^{-1}$yr$^{-1}$ | EMEP/EEA (2019) |
| $N_{\text{retention frac}}$ | N retention fraction | % | Paustian et al. (2006) |
| $L_{\text{housing}}$ | Housing period | day | EMEP/EEA (2019) |
| $x_{\text{TAN}}$ | Fraction of TAN in N excreted | % | EMEP/EEA (2019) |
| $x_{\text{liq}}$ | Regional manure types | % | adapted from FAO (2018) |
| $EF_{\text{N}_2\text{O (stor)}}, EF_{\text{N}_2\text{ (stor)}}$, $EF_{\text{NO (stor)}}, EF_{\text{NH}_3\text{ (small rum)}}$ | European emission factors. Every EFs for small ruminants | % TAN | EMEP/EEA (2019) |
| $EF_{\text{NH}_3\text{ (indoor)}}$ | NH$_3$ European emission factors | % TAN | Sommer et al. (2019) |

#### 2.1.2.3 Organic applications onto land

This section contains the description of the manure application to soil. $m_{\text{(applic,a)}}$ and $m_{\text{(graz,a)}}$ are the manure remaining after

storage and produced during grazing respectively (description is given in 2.1.2.2). Both are yearly stocks applied daily at a constant rate and during a specific period, driven mainly by environmental conditions described below. This assumption may neglect the actual seasonal patterns in the N application usually defined by local governance in some regions. For instance, as discussed in Van Damme et al. (2022), in Europe, the time of the year when fertilizers can be applied is strongly dependent on local regulations. Synthetic fertilizers are also considered in our representation and follow the same temporal distribution as

the manure coming from the storage.

☐ Manure coming from storage and applied to soil as fertilizer

The manure coming from storage is applied daily at a constant rate for 6 months from the beginning of the vegetation growth, corresponding to the first leaf development depending on the PFT. The intermediate period of application ($L_{\text{application}}$ = 6 months) has been chosen in order to take into account the heterogeneity of the agricultural practices

because the model only represents C3 and C4 crop types within the grid-cell. Moreover, there is a lack of information about N application onto grassland at global scale in the literature. We assume that cropland and grassland PFT receive stored manure with a 2 times higher preference for cropland fractions.

☐ Manure deposited during grazing activity by the ruminants

The manure coming from the grazing activity $m_{\text{(graz,a)}}$ is calculated in Eq. 7 and is assumed to be only deposited on

grassland PFTs by the ruminants. The first day of manure deposition for grazing also corresponds to the beginning of



the vegetation growth. The amount and period of manure deposited during grazing are animal-specific and determined by the fraction of time passed at grazing ($x_{\text{graz,a}}$).

#### 2.1.2.4 The soil-plant-atmosphere-processes leading to the soil emissions

We describe the physical processes in the soil that influence ammonia emissions in this section. A single soil TAN pool ($\text{TAN}_{(\text{soil})}$, $\text{gNm}^{-2}$) is considered. The soil TAN pool is dynamically updated depending on the processes implemented in the model. These processes are described in Zaehle and Friend (2010). The ones corresponding to a creation of $\text{NH}_4^+$ are related to mineralization, N applications, and $\text{NH}_x$ deposition, while the losses include nitrification, leaching and volatilization.

$\text{TAN}_{(\text{soil,aq})}$ corresponds to the ammonium pool $\text{TAN}_{(\text{soil})}$ which is assumed to be diluted in the soil water at a different height in the soil according to the $z_{\text{activity}}$ parameter.

The $z_{\text{activity}}$ parameter is regulated by all TAN sources called *input* (*min* : mineralisation, *dep* : deposition, BNF, *fert* : mineral fertilizer, *manure* : applied and grazed manure) in soil.

$$
\begin{aligned}
z_{\text{activity}} =& (p_{\text{zact\_deep}} \times \text{input}_{\text{min}} + \\
& p_{\text{zact\_deep}} \times \text{input}_{\text{dep}} + p_{\text{zact\_deep}} \times \text{input}_{\text{bnf}} + \\
& p_{\text{zact\_surf}}(\text{fert}) \times \text{input}_{\text{fert}} + p_{\text{zact\_surf}}(\text{manure}) \times \text{input}_{\text{manure}} + \\
& z_{\text{activity}} \times \text{TAN}_{(\text{soil})}) \times \frac{1}{\text{input}_{\text{tot}} + \text{TAN}_{(\text{soil})}}
\end{aligned}
\tag{13}
$$

$\text{input}_{\text{tot}}$ is the total TAN sources in soil. We assume that the fertilization and the application of manure are surface N additions to soil whereas the other sources of TAN (mineralisation, deposition, BNF) are deeply added into soil ($p_{\text{zact\_deep}}$ = 1.0 m).

$p_{\text{zact\_surf}}$ is obtained as described in Riddick et al. (2016) by :

$$
p_{\text{zact\_surf}(\text{manure})} = s_{\text{W}}(m) \times \text{input}_{\text{manure}}/\text{SWC}
$$

$$
p_{\text{zact\_surf}(\text{fert})} = s_{\text{W}}(f) \times \text{input}_{\text{fert}}/\text{SWC}
\tag{14}
$$

with SWC the soil water content computed by ORCHIDEE, $s_{\text{W}}(m)$ the specific water volume of manure ($5.67 \times 10^{-4}$ $\text{m}^3[water]\text{g}[N]^{-1}$ (Sommer and Hutchings, 2001; Riddick et al., 2016)) and $s_{\text{W}}(f)$, the specific water volume of synthetic fertilizers. $s_{\text{W}}(f)$ depends on the soil temperature $T_{\text{g}}$ and is given by United Nations Industrial Development Organization (UNIDO) :

$$
S_{\text{W}}(f) = \frac{1 \times 10^{-6}}{0.466 \times 0.66 \times e^{0.0239 \times (T_{\text{g}} - 273)}}
\tag{15}
$$

The emissions of $\text{NH}_3$ ($E_{\text{NH}_3}$, $\text{gNm}^{-2}\,\text{s}^{-1}$) are obtained following the resistive scheme used in the FAN model (Riddick et al., 2016; Vira et al., 2019).

$$
E_{\text{NH}_3} = \frac{\text{NH}_3(\text{g}) - \chi_{\text{a}}}{R_{\text{a}}(z) + R_{\text{b}}}
\tag{16}
$$



with $NH_3(g)$ the $NH_3$ concentration at the surface ($gNm^{-3}$), $\chi_a$, the free-atmosphere concentration ($gNm^{-3}$), $R_a(z)$, the aerodynamical resistance ($sm^{-1}$) and $R_b$, the quasi-boundary layer resistance ($sm^{-1}$).

$\chi_a$ is prescribed as a monthly field averaged over 11 years from a run of the global LMDZ-INCA model at $2.5°x1.3°$

resolution (39 vertical levels) over the 2005-2015 period (Hauglustaine et al., 2014). The spatial distribution of $\chi_a$ is presented in Fig S7. (Supplementary Material) for both May and December (2005-2015 climatology).

$R_a(z)$ is computed interactively by the biophysical module of the ORCHIDEE model. $R_b$ has been implemented according to Xu et al. (2019) as followed :

$$R_b = \frac{v}{D_{NH_3}} \times \left[ \frac{c}{(LAI)^2} \times \left( \frac{l \times \mu_*}{v} \right) \right]^{1/3} \tag{17}$$

with $D_{NH_3}$ the molecular diffusivity of $NH_3$ in air ($m^2s^{-1}$, (Massman, 1998)), c an empirical constant equals to 3, l the leaf width (0.02 m, Massad et al. (2010)), v the kinematic viscosity of air ($1.56 \times 10^{-5}$ $m^2s^{-1}$ at 25°C ), T the air temperature in $K$ and LAI, the Leaf Area Index ($m^2m^{-2}$) which is computed by the ORCHIDEE model. The resulting annual mean $R_b$ ranges between 0 to $1.14$ $sm^{-1}$ over the globe. $D_{NH_3}$ is a function of temperature and is written as:

$$D_{NH_3} = 0.1978 \times \left( \frac{T}{273.13} \right)^{1.81} \times 10^{-4} \tag{18}$$

The Henry's law coefficient ($K_H$) and the dissociation constant of $NH_4^+(aq)$ in water ($K_{NH_4}$) (Sutton et al., 1994) are used for the speciation between the different TAN species ($NH_3(g), NH_3(aq), NH_4^+(aq) \left( gNm^{-3} \right)$).

$$K_H = \frac{[NH_3(aq)]}{[NH_3(g)]} \tag{19}$$

$$K_{NH_4} = \frac{[H^+][NH_3(aq)]}{[NH_4^+(aq)]} \tag{20}$$

By combining equations 19 and 20, we can compute the gaseous phase of ammonia $NH_3(g)$ which is the fraction that will

be volatilized. $TAN_{(soil,aq)}$ corresponds to the aqueous phase of TAN in the soil, which is modulated by the height of the soil through the $z_{activity}$ parameter.

$$NH_3(g) = \frac{TAN_{(soil,aq)}}{\frac{\theta}{K_{fact}} + \epsilon} \tag{21}$$

$\theta$ is the volumetric soil water content (in $m^3[water]m^{-3}[soil]$) and $\epsilon$ the fraction of air-filled soil volume computed by the ORCHIDEE model.

$K_{fact}$ is calculated with:

$$K_{fact} = 1/(1 + K_H + K_H [H^+]/K_{NH_4}) \tag{22}$$



and $K_{\mathrm{H}}$ the Henry's law constant for $\mathrm{NH}_{(3)}$ depends on the surface temperature $T_{\mathrm{g}}$ :

$$K_{\mathrm{H}} = \mathrm{H} \times T_{\mathrm{g}} \times e^{4092(1/T_{\mathrm{g}}-1/T_{\mathrm{ref}})} \tag{23}$$

with $T_{\mathrm{ref}}$, the reference temperature (298.15 K) and $H$, a conversion factor, equals to 4.905. We use the value of 0.59

$\mathrm{molm}^{-3}\mathrm{Pa}^{-1}$ described in Sander (2015) to which the perfect gas constant has been multiplied in order to get a constant without unit. $K_{\mathrm{NH}_4}$ is the dissociation equilibrium also depending on the surface temperature $T_{\mathrm{g}}$ as follows:

$$K_{\mathrm{NH}_4} = 5.67 \times 10^{-10} \times e^{-6286(1/T_{\mathrm{g}}-1/T_{\mathrm{ref}})} \tag{24}$$

The concentration in hydrogen ion $[\mathrm{H}^+]$ is assumed to be constant and equal to $10^{-7}$ which corresponds approximately to the pH given in Massad et al. (2010) for cattle manure, di-ammonium phosphate fertilizers in acidic soils and ammonium

nitrate fertilisers. A pH of 7 is also adopted in Riddick et al. (2016). In our modeling, the pH does not impact the surrounding soil pH in our model, in contrast to Vira et al. (2019) where the pH varies according to different TAN age classes.

## 2.2  Modeling set-up

The ORCHIDEE model, including all the developments described in Section 2 was run at a spatial resolution of 2 ° (180x90). This spatial resolution is relatively low but enables to perform an ensemble of sensitivity tests at a reasonable computing cost.

We also performed a reference simulation at 0.5° resolution to ensure that the model resolution does not affect the results. We performed a 10-year reference simulation over the 2005-2015 period. This simulation starts in January 2005 from a simulation done with an ORCHIDEE version similar to the one presented in this paper but without the developments presented in Section 2.1.2. In the reference simulation, all annual forcing data are updated every year, except those related to BNF and livestock density constant over time. A set of 9 sensitivity test simulations characterized by specific changes in the parametrization was

conducted to evaluate the impact of parameters uncertainty on agricultural ammonia emissions. The parameters that have been tested are the atmospheric ammonia concentration ($\chi_{\mathrm{a}}$), the pH of the manure (pH, default value : 7), the timing period of the N application ($L_{\mathrm{application}}$, default value : 183 days), the emission factor for the housing and storage activities ($\mathrm{EF}_{\mathrm{NH}_3\text{ (indoor)}}$), the fraction of ammonium in the fertiliser ($\mathrm{Frac}_{\mathrm{NH4^+,fert}}$, default value : 0.6) and the N deep processes regulation parameter ($p_{\mathrm{zact\_deep}}$, default value : 1m). Table 5 summarizes the set of simulations with the key parameters tested.


In Riddick et al. (2016), the value of $\chi_{\mathrm{a}}$ was fixed to $0.3\mu\mathrm{gNm}^{-3}$ as it is representative of the concentration over low-activity agricultural sites (Zbieranowski and Aherne, 2012). Little sensitivity of the emissions to this parameter was found since $\chi_{\mathrm{a}}$ is much smaller than $\mathrm{NH}_3(\mathrm{g})$. However, this parameter has been tested in our implementation through a sensibility analysis.

The ORCHIDEE model requires a set of forcing data which are described hereafter:

☐ Meteorological data includes near-surface air temperature and specific humidity, wind speed, pressure, short- and long-wave incoming radiation, rainfall, and snowfall. This information comes from the CRU-JRA V2.1 dataset (Harris et al.,





**Table 5.** Summary of the simulations performed with the parameters tested. $EF_{NH_3 \text{ (indoor)}}$ are the one described in section 2.1.2.2 for housing and storage emissions.

| Simulation | Parameter tested | value |
|---|---|---|
| $CONC_{0.3}$ | $\chi_a$ | $0.3\ \mu gNm^{-3}$ |
| $CONC_3$ | $\chi_a$ | $3\mu gNm^{-3}$ |
| $pH_{7.5}$ | pH | 7.5 |
| $TIM_{10}$ | $L_{application}$ | 10 days |
| $TIM_{365}$ | $L_{application}$ | 365 days |
| $EF_{max}$ | $EF_{NH_3 \text{ (indoor)}}$ | (ref value + standard deviation) see Table 3 |
| $FERT_{0.75}$ | $Frac_{NH4^+,fert}$ | 0.75 |
| $p_{zact\_deep,1.5}$ | $p_{zact\_deep}$ | 1.5 m |

2014) (pre-processed and adapted by V. Bastrikov, LSCE, July 2020). The data provides meteorological information at 6-hourly time step;

☐ Global average annual atmospheric $CO_2$ concentration which is provided by (TRENDY, Le Quéré et al. (2018));

☐ Global annual land-cover distribution based on the combined information from the LUH2v2 dataset at 0.25° resolution (Hurtt et al., 2020) and the ESA CCI Land Cover (see Lurton et al. (2020) for more details);

☐ Atmospheric N deposition fluxes ($NH_x$ and $NO_y$) are taken from the IGAC/SPARC Chemistry-Climate Model Initiative (CCMI, Eyring et al. (2013)) and have been used in the NMIP project (Tian et al., 2018). They correspond to information at 0.5° resolution, at a monthly time resolution;

☐ The mineral fertilizer annual rates over croplands and grasslands come from an annual dataset developed by (Lu and Tian, 2017). It corresponds to a reconstruction from 1960 to 2014 for the global cropland, matched with HYDE 3.2 cropland distribution;

☐ Biological nitrogen fixation rate is provided as a climatological data as a function of the evapotranspiration flux (see Vuichard et al. (2019) for more details);

☐ The distribution of each livestock category is taken from the Gridded Livestock of the World (GLW 2 (Robinson et al., 2014)) representing the gridded animal densities $D_a$ for the year 2006 at 1 km resolution. Small ruminant densities correspond to the sum of the cheap and goat densities. Dairy cattle distribution has been retrieved from the total cattle distribution combined with national dairy cattle densities given by FAOSTAT (2020). The calculation adopted is described in the Supplement.





## 2.3 Model evaluation dataset

Our integrated approach allows the computation of different variable levels before the final emission results, such as biomass productivity, animal excretion rate, and manure production. This set of variables offers the advantage of evaluating our emissions at different stages of the nitrogen flow with previous works listed in Table 6.

**Table 6.** Summary of the different simulated variables which are evaluated in this work by comparison with previous studies.

| Metric | Description | Unit | Sources of previous studies |
|---|---|---|---|
| $BM_{crop/grass}$ | Global crop and grass production | $TgN yr^{-1}$ | Bouwman et al. (2013a) <br> Billen et al. (2014); Bodirsky et al. (2014); <br> Conijn et al. (2018); Uwizeye et al. (2020) |
| $BM_{ing,tot}$ | Global N intake by livestock | $TgN yr^{-1}$ | Billen et al. (2014); Bodirsky et al. (2014); <br> Conijn et al. (2018); Uwizeye et al. (2020) |
| ER | Excretion rate depending on the animal type | $kgN(1000\,kg_{animal\,mass}^{-1})\,day^{-1}$ | Paustian et al. (2006) |
| $E_{NH_3\,(indoor)}$ | NH$_3$ emissions from house, yard and storage | $kgN m^{-2} yr^{-1}$ | Crippa et al. (2018); Vira et al. (2019) |
| $m_{N,excr/applic}$ | Global amount of manure excreted / applied | $TgN yr^{-1}$ | Beusen et al. (2008); Potter et al. (2010); <br> Bouwman et al. (2013b); Billen et al. (2014); <br> Bodirsky et al. (2014); Zhang et al. (2017a); <br> Conijn et al. (2018); Vira et al. (2019); <br> Uwizeye et al. (2020) |
| $F_N$ | Global amount of fertilizer applied | $TgN yr^{-1}$ | Bouwman et al. (2013b); Billen et al. (2014) <br> Bodirsky et al. (2014); Zhang et al. (2017a); <br> Conijn et al. (2018); Vira et al. (2019); <br> Uwizeye et al. (2020) |
| $E_{NH_3}$ | NH$_3$ emissions from agriculture | $TgN yr^{-1}$ | Hoesly et al. (2018); Vira et al. (2019) <br> Evangeliou et al. (2020) |

Decadal mean (2005-2015) value of global and regional calculated agricultural emissions, including indoor and soil emis-
sions, are compared to the CEDS inventory (Hoesly et al., 2018) and the emissions simulated by FANv2 (Vira et al., 2019).
From the LMDZ-OR-INCA coupling development perspective, it is interesting to compare our approach with CEDS (Hoesly
et al., 2018) as it is a reference dataset offering a long period of data (1750-2019). FANv2 has been chosen for our evalu-
ation since our work is based on a similar approach. The regional budget account for Africa, Asia Tropical South, Europe,
China-Korea-Japan (abbreviated as China-K-J in the Figures), Oceania, India, USA-Canada, and Latin America. The sea-
sonal variations of ammonia emissions are also evaluated against satellite-derived emissions (Evangeliou et al., 2020). For that
purpose, atmospheric NH$_3$ columns observed by the IASI satellite have been combined with the NH$_3$ lifetime calculated by
LMDZ-INCA in order to retrieve emissions. The NH$_3$ retrieval product used to derive emissions in our study is the 2011-





2015 morning observations (Metop A and B) and follows a neural network retrieval approach (ANNI-NH3-v3R) as referred in Van Damme et al. (2017, 2021). Both the lifetime and atmospheric columns are monthly products and share the exact grid res-
olution (LMDZ-INCA grid resolution at 2.5°x1.3°). All three CAMEO, CEDS, and FANv2 seasonal variations are evaluated against IASI-derived emissions (defined as IASI$^{inv}$). In order to be consistent with IASI observations (where no distinction in the sources are possible), CAMEO, CEDS and FANv2 agricultural emissions need to be complemented by the fire emission data taken from van der Werf et al. (2017) (GFED s4), and by industrial and waste sources (Hoesly et al., 2018). The extended emissions are referenced to as CAMEO$_+$, FAN$_+$ and CEDS$_+$, as described in Table 7. It is important to note that only the
ORCHIDEE model can provide natural emissions; this source is not considered in the FAN$_+$ and the CEDS$_+$ dataset.

**Table 7.** Summary of the different dataset used in the comparison seasonality analysis with the IASI-derived emissions. All the emission sets (excepting FANv2 data which is a 2010-2015 climatology) are taken from 2011-2015 period and have been gridded onto the LMDZ-INCA default resolution 144x142.

| Configuration | Emission category | Data sources |
|---|---|---|
| CAMEO$_+$ | Agricultural emissions | ORCHIDEE run |
| | Natural emissions | ORCHIDEE run |
| | Waste and industrial sources | CEDS (Hoesly et al., 2018) |
| | Biomass burning | GFEDs4 (van der Werf et al., 2017) |
| FAN$_+$ | Agricultural emissions | FANv2 data (2010-2015) (Vira et al., 2019) |
| | Natural emissions | Not taken into account in this dataset |
| | Waste and industrial sources | CEDS (Hoesly et al., 2018) |
| | Biomass burning | GFEDs4 (van der Werf et al., 2017) |
| CEDS$_+$ | Agricultural sources | CEDS (Hoesly et al., 2018) |
| | Natural emissions | Not taken into account in this dataset |
| | Waste and industrial sources | CEDS (Hoesly et al., 2018) |
| | Biomass burning | GFEDs4 (van der Werf et al., 2017) |

## 3   Results and evaluation

### 3.1   Evaluation of intermediate variables

Using the ORCHIDEE model, we estimate a total biomass produced, including grass and crop, of about 103TgNyr$^{-1}$, slightly lower than previous estimates (110-152TgNyr$^{-1}$ estimated by Bouwman et al. (2013a); Billen et al. (2014); Bodirsky et al.
(2014); Conijn et al. (2018); Uwizeye et al. (2020)). The calculated global annual crop production (expressed in N) is about 74TgNyr$^{-1}$ and compares well with 72 and 74TgNyr$^{-1}$ estimated by Billen et al. (2014) and Zhang et al. (2021). The calculated annual grass production (25.5TgNyr$^{-1}$) is more than 3 times lower than the 80.3TgNyr$^{-1}$ reported by Billen et al. (2014) and estimated from the difference between livestock ingestion and available feed resources. Our resulting total biomass





ingested by the livestock ($88 \mathrm{TgNyr}^{-1}$) is lower than the range found in the literature ($122\text{-}167 \mathrm{TgNyr}^{-1}$, Billen et al. (2014);
Bodirsky et al. (2014); Conijn et al. (2018); Uwizeye et al. (2020) ) which can be attributed to the low grassland production
calculated in our model. The excretion rate computed by our model is also 3 times lower than the values given by Paustian
et al. (2006) meaning that the nitrogen excreted by the animal is low regardless of the available biomass. It can be explained by
using a unique grass C:N ratio per pixel and a global C:N ratio for crop in the model. However, Paustian et al. (2006) reported
a $50\%$ uncertainty on these coefficients. In our calculation, the manure produced is directly applied to the soil. Global annual
amount of manure production ($66 \mathrm{TgNyr}^{-1}$) is lower than the range of $99\text{-}129 \mathrm{~TgNyr}^{-1}$ estimated in recent studies (Beusen
et al., 2008; Potter et al., 2010; Bouwman et al., 2013b; Billen et al., 2014; Zhang et al., 2017a; Conijn et al., 2018; Vira et al.,
2019; Uwizeye et al., 2020).

Figure 2 compares the distribution of the N manure applied with the values retrieved by Zhang et al. (2017a) for year 2006
in order to be consistent with the reference livestock distribution used in our approach. The spatial distribution of the manure
application highlights the main livestock-raising regions such as China, India, Europe, Latin America, and the USA. It shows
good consistency with Zhang et al. (2017a) although it is higher in India, the USA, and Latin America and lower in China and
Europe. These differences can be explained by the fact that we use different regional animal weights per livestock category
(Table 1) instead of a fixed value in Zhang et al. (2017a). It induces different N demands for similar livestock and ultimately
different quantities of applied manure. Indeed, we use recently adapted data for animal weights from FAO (2018) while Zhang
et al. (2017a) use IPCC guidelines (Tier1 IPCC 2006; Paustian et al. (2006)). For instance, for India, non-dairy cattle weight
is almost 4 times higher, which explains the differences observed in our calculation. Moreover, our study assumes a unique
nitrogen excretion rate per livestock type and no livestock system distinctions as a simplification.

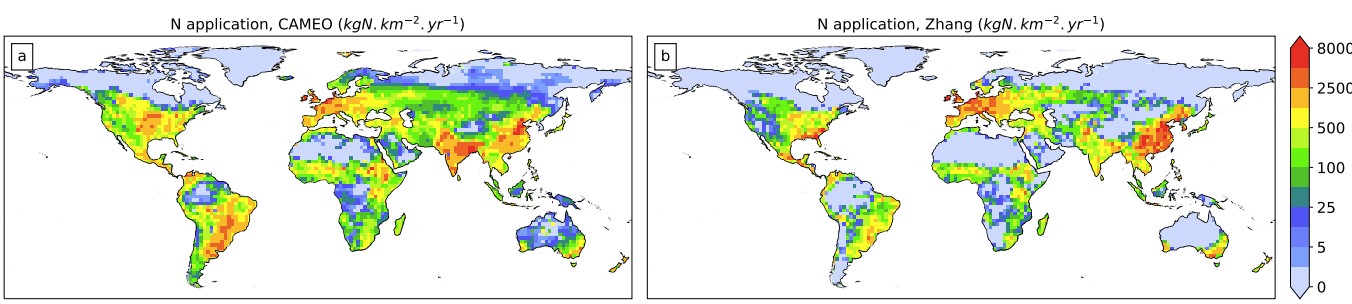

**Figure 2.** Manure application ($\mathrm{kgNkm}^{-2}\mathrm{yr}^{-1}$) simulated by CAMEO and averaged over 2005-2015 (a) and calculated by Zhang et al.
(2017a) for 2006 (b).

## 3.2 Agricultural emissions at the global scale

We estimate global $\mathrm{NH}_3$ agricultural emissions (average over 2005-2015) of about $44 \mathrm{TgNyr}^{-1}$ of which $78\%$ comes from soil
volatilization (driven by fertilizer and manure applications) and the remaining from indoor emissions (from livestock housing,



**Table 8.** Global estimates of intermediate variables computed by our model for 2005-2015 period and the range of previous estimates.

| Metric | This study | Range of previous estimates |
|---|---|---|
| $\mathrm{NPP_{crop/grass}}$ (TgNyr$^{-1}$) | 103 | 110 - 152 |
| $\mathrm{BM_{ing,tot}}$ (TgNyr$^{-1}$) | 88 | 122 - 167 |
| ER (kgN$(1000\,\mathrm{kg_{animal\,mass}^{-1}})$day$^{-1}$) | 0.21 - 0.35 | 0.31 - 1.47 |
| $m_{\mathrm{N,excr/applic}}$ (TgNyr$^{-1}$) | appl : 66 | excr : 99.9 - 129 , appl : 32 - 131 |
| $F_{\mathrm{N}}$ (TgNyr$^{-1}$) | 121.6 | 55 - 116 |

yard, and storage). These global NH$_3$ emissions are within the range given by Hoesly et al. (2018) and Vira et al. (2019) (39 - 47TgNyr$^{-1}$) (Fig 3.a).

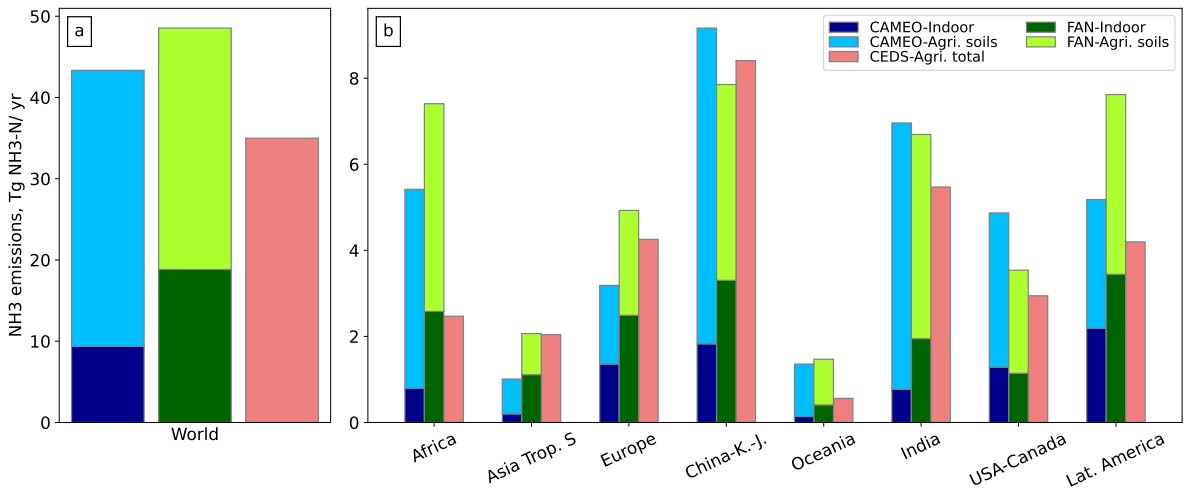

**Figure 3.** Averaged global (a) and regional (b) NH$_3$ emissions (TgNyr$^{-1}$) from indoor activities ('Indoor' in darker bars) and agricultural soil ('Agri. soils' in lighter bars) computed by CAMEO over 2005-2015 ('CAMEO' in blue bars) and FANv2 (Vira et al., 2019) over 2010-2015 ('FAN' in green bars). Total agricultural emissions (accounting for manure management, soil volatilization) estimated in the CEDS inventory (2005-2014 average) (Hoesly et al., 2018) are represented in purple ('CEDS-Agri. total'). China-K-J accounts for China-Korea-Japan

China, India, Africa, Latin America, the USA, and Europe appear as the main contributors to the global NH$_3$ emissions accounting for 80% of the total budget (Fig 3.b). Most of these source areas, which have also been identified as agricultural regions by Van Damme et al. (2018), are regions with intensive crop cultivation (Fig S5. of the Supplement) and important livestock activities, inducing high N application rates (Fig 2). Spatial distributions of the calculated agricultural NH$_3$ emissions shows a good agreement with FANv2 and CEDS results (Fig 4, b and c). In India and China, our emissions are slightly higher than FANv2 and CEDS estimates. They are lower than the FANv2 estimate in Latin America and Africa but high compared






to CEDS emissions, particularly low in these two regions. In some parts of Africa and Latin America, where the use of synthetic fertilizer is low (never exceeding $2500 \text{kgNkm}^{-2}\text{yr}^{-1}$), the livestock activity appears to be the main contributor to the emissions.

In intensive agricultural regions, data used for mineral fertilizer application rates can be a source of discrepancy between models. Vira et al. (2019) use the Landuse Harmonization 2 dataset (Hurtt et al., 2020) which assumes that only croplands are fertilized. The amounts of fertilizer applied over croplands are comparable globally between Vira et al. (2019) and our study (respectively min-max: $79\text{-}87\text{TgNyr}^{-1}$ and $96\text{-}101\text{TgNyr}^{-1}$ over 2010-2015) but differ in some regions (see Fig **??**.a). In addition, in our study, grasslands are also fertilized with a global amount of $25.7\text{TgNyr}^{-1}$. It leads to differences in the simulated soil emissions, more specifically in India, the USA, and China, where grasslands are highly fertilized (Fig **??**.b) and can be translated into high volatilization rates when compared to FANv2.

$NH_3$ emissions peak in June-July-August for most regions (the USA, Europe, China and Africa, Fig 5) with maximum values reaching $16.4\text{gNm}^{-2}\text{yr}^{-1}$ in Eastern China. The peak in India and Latin America appears somewhat earlier during spring or SON and DJF, respectively. Depending on the region, the seasonality of the emissions varies according to different factors, including environmental parameters and agricultural practices. This aspect will be analyzed in more detail in Sections 3.4 and 3.5.

The spatial pattern of the simulated indoor $NH_3$ emissions (Fig 6 .b) is similar to that of manure application rates, being both driven mainly by livestock density. Hotspot regions of indoor emissions are located in Eastern China, Eastern India and Northern Europe, with maximum values reaching up to $1.7\text{gNm}^{-2}\text{yr}^{-1}$. The major sources of volatilization from soils are located in India, Eastern China and the USA with maximum value of $12\text{gNm}^{-2}\text{yr}^{-1}$. The difference in spatial patterns between the two source categories is due mainly to the fact that soil emissions not only depend on livestock distribution - indoor emissions - but also environmental conditions and mineral fertilizer application rates. The sensitivity of modeled emissions to some of these factors is presented in Section 3.4.

The agricultural emissions from manure management (qualified as 'indoor') are poorly quantified at the global scale. While Vira et al. (2019) evaluated this emission source to $18\text{TgNyr}^{-1}$ for 2010, our estimate is twice lower ($9.6\text{TgNyr}^{-1}$) but is in good agreement with the $NH_3$ emissions reported by Crippa et al. (2018) and Beusen et al. (2008) ($9\text{TgNyr}^{-1}$ for year 2010 and 2000 respectively). Biomass excreted in our model is 40% lower than what is produced in FANv2, which can partly explain the difference observed in the resulting indoor emissions. In addition, we use EF from recent studies (Sommer et al., 2019; EMEP/EEA, 2019) while a parametrization relying on the temperature and the ventilation rate is used in FANv2 (Vira et al., 2019). However, the parametrization in FANv2 has been adjusted to reproduce default EFs for barns and stores from EMEP/EEA (2016) under European conditions. The use of updated EF compared to Vira et al. (2019) largely explains the differences between the estimated indoor emission estimations. In addition, in contrast to FANv2, our manure management module integrates a distinction between solid and liquid manure handling for each livestock type, with very different EF values. As discussed in Groen et al. (2016); Mu et al. (2017) and Uwizeye et al. (2017, 2020), uncertainties associated with EFs are large and can lead to over- or under-estimates in indoor emissions and resulting soil emissions. The sensitivity of our calculated total emissions (manure management and soil) to this input parameter will be described in detail in Section 3.4, with





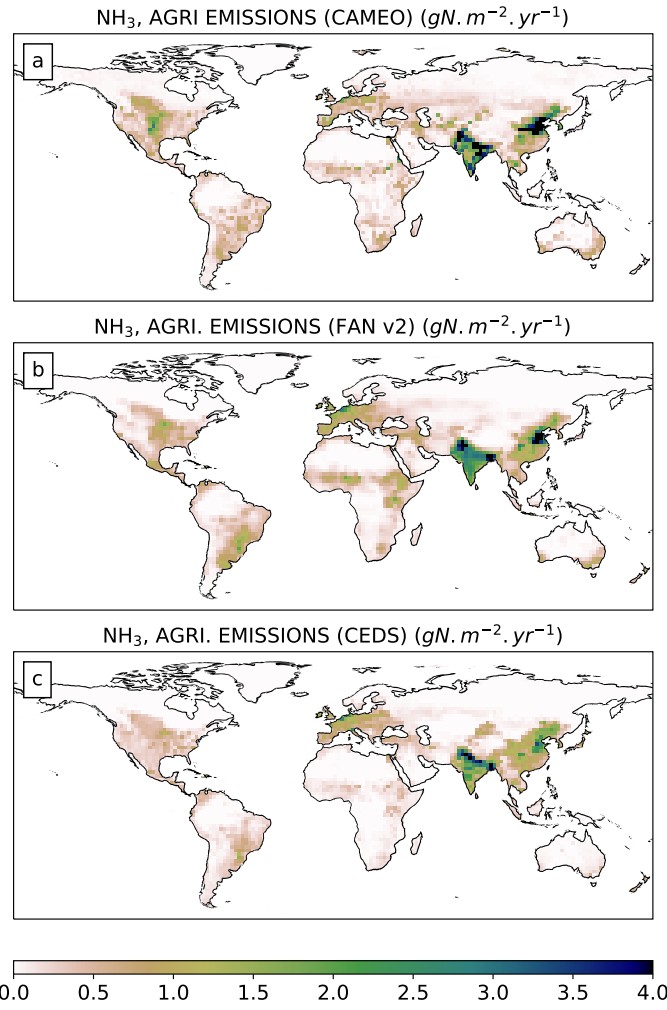

**Figure 4.** Simulated ammonia emissions (gNm$^{-2}$yr$^{-1}$) from total agricultural sources computed (a) by CAMEO (2005–2015 average), (b) by the FANv2 model (2010-2015 average) (Vira et al., 2019) and (c) from the CEDS inventory (2005-2015 average) (Hoesly et al., 2018)

a change of 14% at the global scale, demonstrating that the parameter has a significant impact on indoor emissions. Moreover,
445  we consider that each animal category has unique grazing, housing, and yard periods while Vira et al. (2019) consider regional livestock production systems.

### 3.3 Emissions at the regional scale

A good agreement is found for the NH$_3$ emissions in China between CAMEO, CEDS, and FAN estimates. In agreement with CEDS, India is the second biggest emitter region (Fig 3.b). However, FANv2 estimates much more emissions in Africa and



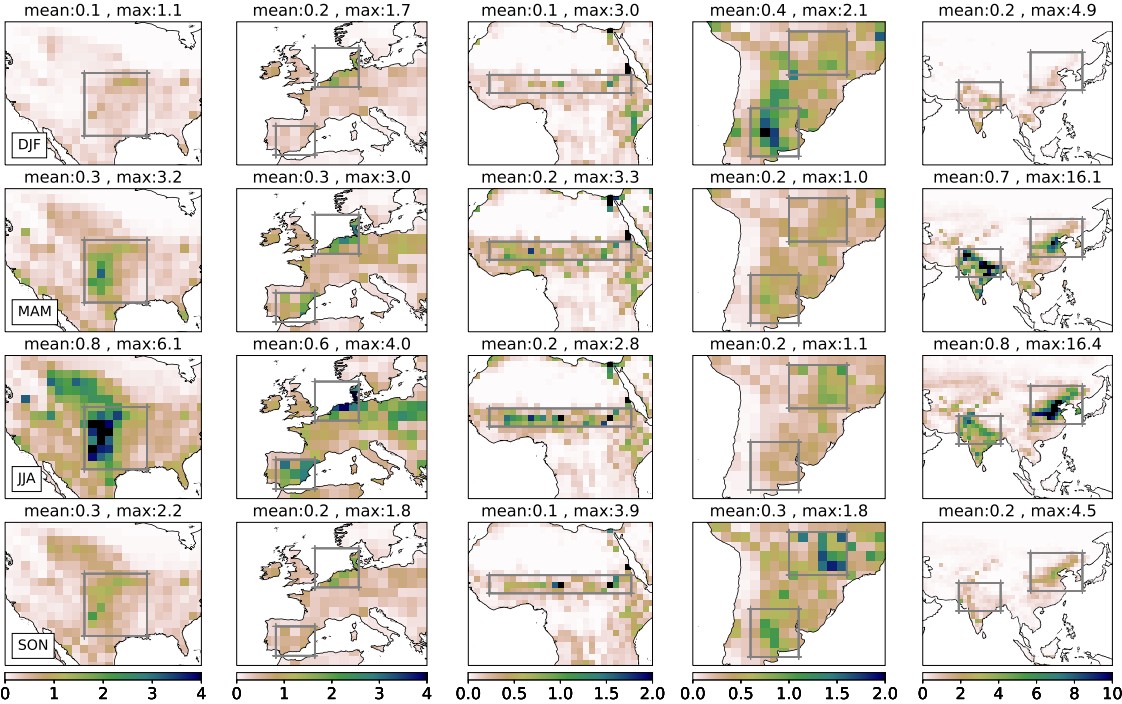

**Figure 5.** Regional seasonal agricultural ammonia emissions averaged over 2005–2015 $(\mathrm{gNm^{-2}yr^{-1}})$ simulated by CAMEO. Boxes delimit the regions used in the analysis comparing CAMEO emissions with IASI[inv] (Section 3.5). Please note that the scales are different for each region.

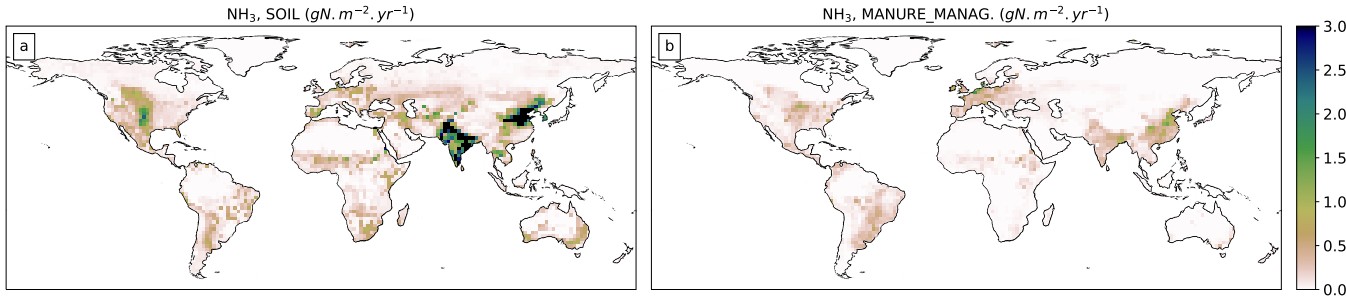

**Figure 6.** Simulated ammonia emissions averaged over 2005–2015 $(\mathrm{gNm^{-2}yr^{-1}})$ from agricultural soil (fertilizer and manure application) (a) and manure management (b).

450  Latin America. As mentioned, there are important gaps between estimates given by CEDS and FANv2 for these two regions. In Africa and Latin America, our calculation leads to intermediate results between CEDS and FANv2 estimates.



Our estimates are quite similar to those given by FANv2 and CEDS, especially in Europe, China, and India. However, our indoor emissions are usually lower than what is computed by FANv2, except in the USA, where our emissions are slightly higher. In China, our total agricultural estimate of $9.3\,\mathrm{TgNyr^{-1}}$ is higher than the ones from CEDS and FANv2. However, our estimate is lower than those from several studies focusing on Chinese emissions: Kang et al. (2016), Li et al. (2021), Zhang et al. (2018); Crippa et al. (2018) and Zhang et al. (2017b) (respectively 9.7, 10.0, 10.3, 11.3, $12.4\,\mathrm{TgNyr^{-1}}$). In India, the emissions we compute ($7.1\,\mathrm{TgNyr^{-1}}$) are closed to the FANv2 emissions ($7.4\,\mathrm{TgNyr^{-1}}$). Our emissions in North America are 30% higher than the FANv2 one and the other estimates. As shown in Fig 6 South Central part of the USA (mainly Texas) is the hotspot of the region where the maximum value can reach $3\,\mathrm{gNm^{-2}yr^{-1}}$. The combination of low soil moisture and a high temperature simulated in this area can explain such high values of volatilization from the soil. In FANv2, emissions do not exceed $1.5\,\mathrm{gNm^{-2}yr^{-1}}$. Unlike the USA, our emissions in Europe are 30% lower than FANv2 and EDGAR4.3 emissions and 15% lower than EMEP and CEDS ones.

Unlike in FANv2, where three types of N fertilizers in the form of ammoniacal nitrogen, urea, and nitrates are considered, we assume a constant ammonium fraction of 0.6 in synthetic fertilizers. Even if the yearly fertilizer application is similar to the amount used in FANv2, the ammonium pool in soil from the mineral application can be different. It may imply differences in the emissions, especially in regions where the mineral application is intensive such as Europe, China, and India (See Fig S5. from the Supplement Material). Concerning Africa and Latin America, our calculated emissions (~$5.4\,\mathrm{TgNyr^{-1}}$) are within the range of CEDS and FANv2. Africa and Latin America are characterized by specific environmental conditions along with different vegetation types, which may explain the uncertainties present in the estimates. There is a lack of information regarding agricultural practices and resulting emissions in these regions. In Argentina, Castesana et al. (2018) estimated agricultural emissions of about 0.31 TgN while our emissions reach 0.91 TgN and are closer to Vira et al. (2019) estimates (1.02 TgN). The large differences mainly come from fertilizer use, reaching 1400 Gg N in their approach. The fertilizer use from NMIP project Tian et al. (2018) (752 Gg N) is in line with the reported values in Castesana et al. (2018) and are consistent with the IFA statistics for 2010-2015 (400-900 GgN). We can not easily conclude whether the emissions differences come from emission factors or manure production estimation. We can only compute a posteriori single emission factor for soil emission from our process-based modeling, while no manure stock production is given in Castesana et al. (2018).

### 3.4 Sensitivity to model parameters

Among the parameters tested, the pH used in the calculation of the gaseous phase of ammonia is the strongest driver of $NH_3$ emissions (Fig 7). At the global scale, the pH induces an increase of about 74% when fixed at 7.5 compared to the reference value fixed at 7.

The impact of the pH is very variable from one region to another and reaches up to 90% in some regions such as Africa and the USA, while it is the lowest in India (49%) (Fig 7). In order to explain these regional differences, we explored the drivers of the spatial distribution of modeled $NH_3$ emission sensitivity to pH. The spatial distribution of the sensitivity to pH of $NH_3$ emissions and the gaseous ammonia pool of soil are similar (Fig 8.a). In particular, the sensitivity is low in India compared to other regions like Europe for both variables. Fig 8.b shows the spatial pattern of the dissociation constant of ammonium.



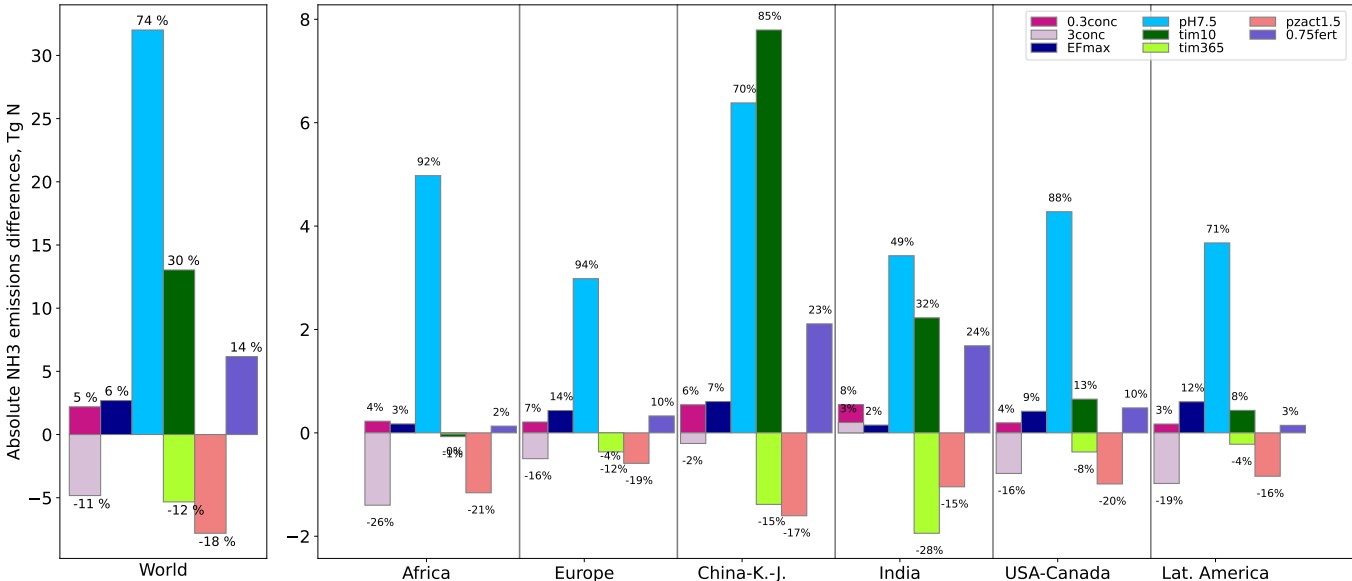

**Figure 7.** Global and regional differences between $NH_3$ emissions from the test simulation (TEST : $CONC_{0.3}$, $CONC_3$, $pH_{7.5}$, $TIM_{10}$, $TIM_{365}$, $EF_{max}$, $pzact_{1.5}$, $FERT_{0.75}$) and the reference simulation (REF) in TgN. Percentages indicate the change relatively to the REF value $(TEST - REF)/REF \times 100$. China-K-J accounts for China-Korea-Japan.

The highest values are located in the warmest regions, such as India, where the temperature is one of the main drivers of the dissociation constant. As the dissociation reaction (Eq 20) is favored in these regions (more $NH_3$ is available), it implies that volatilization is more likely to occur. Along with the high dissociation constant, India is characterized by important soil $NH_4^+$ concentration (Fig 8.c) due to intensive agricultural input (mineral fertilizer and manure applications), leading to a high
quantity of TAN available for emissions. In regions where conditions promote high $NH_3$ volatilization, pH is a weaker driver of emissions. Despite the regional differences in the pH sensitivity, it is an environmental parameter that is an essential driver in the emissions and can be a source of significant uncertainties in our model.

Riddick et al. (2016) also present the results of the emission sensitivity to pH change. They estimate an increase of 50% and 70% in the manure and fertilizer emissions, respectively, when changing pH from 7 to 8. Even though the sensitivity we
describe seems higher than in Riddick et al. (2016), we can hardly conclude since we consider a unique pool of TAN. Indeed, we calculate the impact of a change in the total emissions, whereas in Riddick et al. (2016), both changes in the manure and fertilizer emissions are calculated by changing the pH of the 2 TAN pools (manure and the fertilizer) separately.

Changing the duration of the N application (mineral fertilizer and stored manure) from 183 days to 10 days induces a 30% increase of global $NH_3$ emissions. The highest increase is calculated in China (86%), while Europe's impact is slightly
negative (-4%). Reducing the duration of fertilization induces a significant change in the emission dynamic (Fig 9), with emission peaks occurring right after the start of the growing vegetation season, considered in our model as the first day of



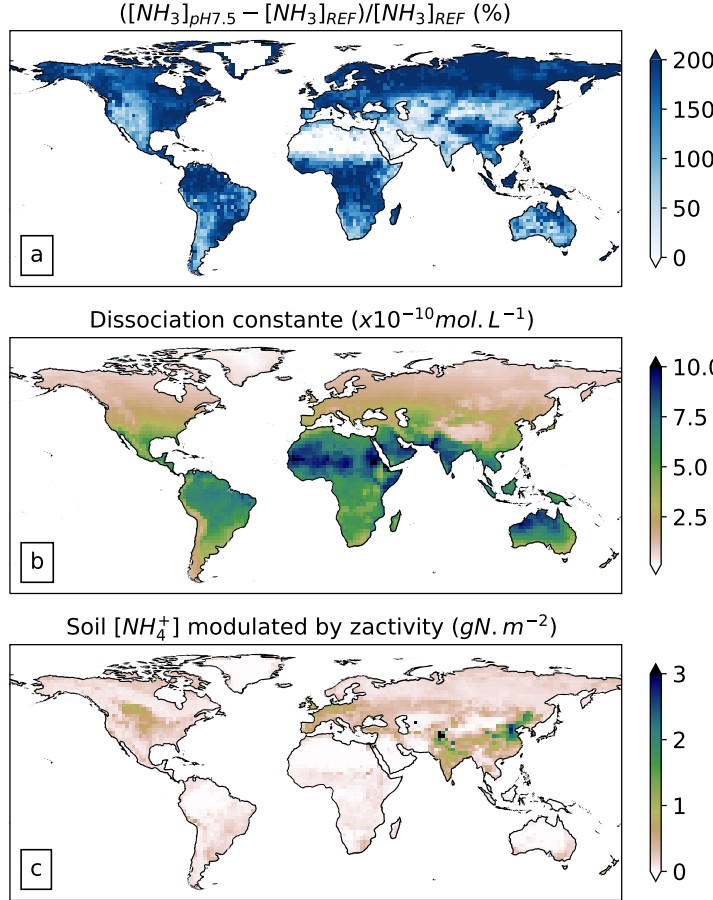

**Figure 8.** Relative anomaly of gaseous ammonia in soil between the pH7.5 simulation and the reference simulation (a) in %. $NH_4^+(aq)/NH_3(aq)$ dissociation constant from the reference simulation (b) in $molL^{-1}$. The soil $NH_4$ concentration modulated by the zactivity parameter (c) in $gNm^{-2}$.

.

the N application period (Fig 10). However, the emission sensitivity to fertilization duration varies across regions, depending on the environmental conditions after the start of the vegetation season. In China, this signal is on average higher in April-May (Fig 10). It is the period with the lowest soil moisture value and the highest soil temperature, conditions that maximize emissions. It could explain the high sensitivity we observe in this region. On the contrary, in Europe, the growing season signal



appears mainly in February and April, where the soil temperature is the lowest and the soil moisture the highest, indicating that conditions are the least favorable to emissions, resulting in a negative sensitivity.

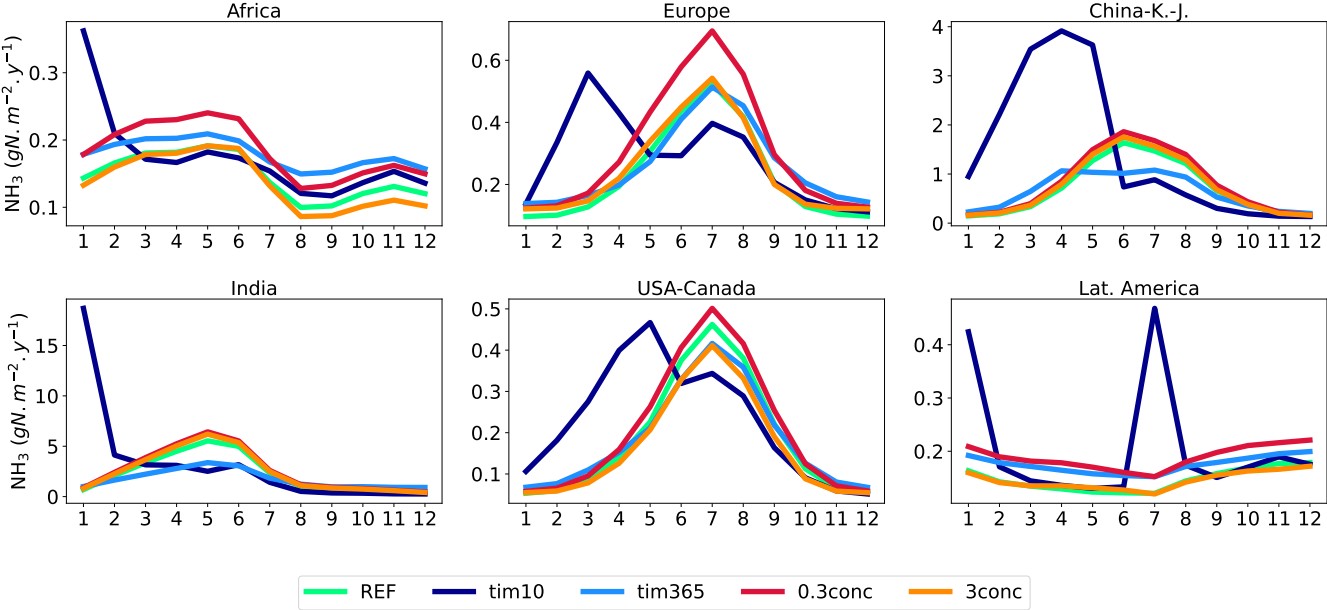

**Figure 9.** Regional $NH_3$ emissions ($gNm^{-2}yr^{-1}$) from different set of simulation. Green lines are the emissions from the reference CAMEO simulations. Dark and light blue lines are emissions respectively from the $TIM_{10}$, $TIM_{365}$ simulations. Dark red and orange lines are emissions respectively from $CONC_{0.3}$, $CONC_3$ simulations. China-K-J accounts for China-Korea-Japan

When N is constantly applied during the whole year (365 days), the emissions are reduced by about 12% globally, with India being the region with the strongest reduction (-28%). The emissions are lower when N is applied the whole year since it

reduces the quantity of N emitted when conditions are the most favorable for volatilization. The variation of $p_{zact\_deep}$ from 1m to 1.5m has a relatively constant impact on $NH_3$ emissions of about -20% over every region. Increasing $p_{zact\_deep}$ increases the dilution of specific ammonium sources (that were assumed as 'deep sources': BNF, deposition, and mineralization) in soil, which in turn reduces emissions. Concerning the sensitivity to the content of ammonium in N fertilizers, when $Frac_{NH4+,fert}$ is increased by 20%, emissions increase by about 14% on average. In China and India, where fertilizer use is the highest, the

increase can reach +24% (see Fig S5). When fixing the atmospheric concentration at $0.3\mu gNm^{-3}$ and $3\mu gNm^{-3}$ the global $NH_3$ emissions increase by 5% and decrease by 11%, respectively. In Africa, where the impact of using a concentration of 3 [$\mu gNm^{-3}$ is the highest, emissions are reduced by 26%, while in India, they slightly increase (3%). Indeed, over India, atmospheric $NH_3$ concentrations from LMDZ-INCA are higher than $3\mu gNm^{-3}$, in particular during May with values up to $7\mu gNm^{-3}$ (Fig S7 in the Supplementary Material). Counter-intuitively, using fixed atmospheric $NH_3$ concentration ($CONC_{0.3}$

and $CONC_3$ simulations) do not induce any important change in the seasonality of emissions (Fig 9). The sensitivity to this parameter has also been tested in FANv1, and they found the same range of model response (Riddick et al., 2016).





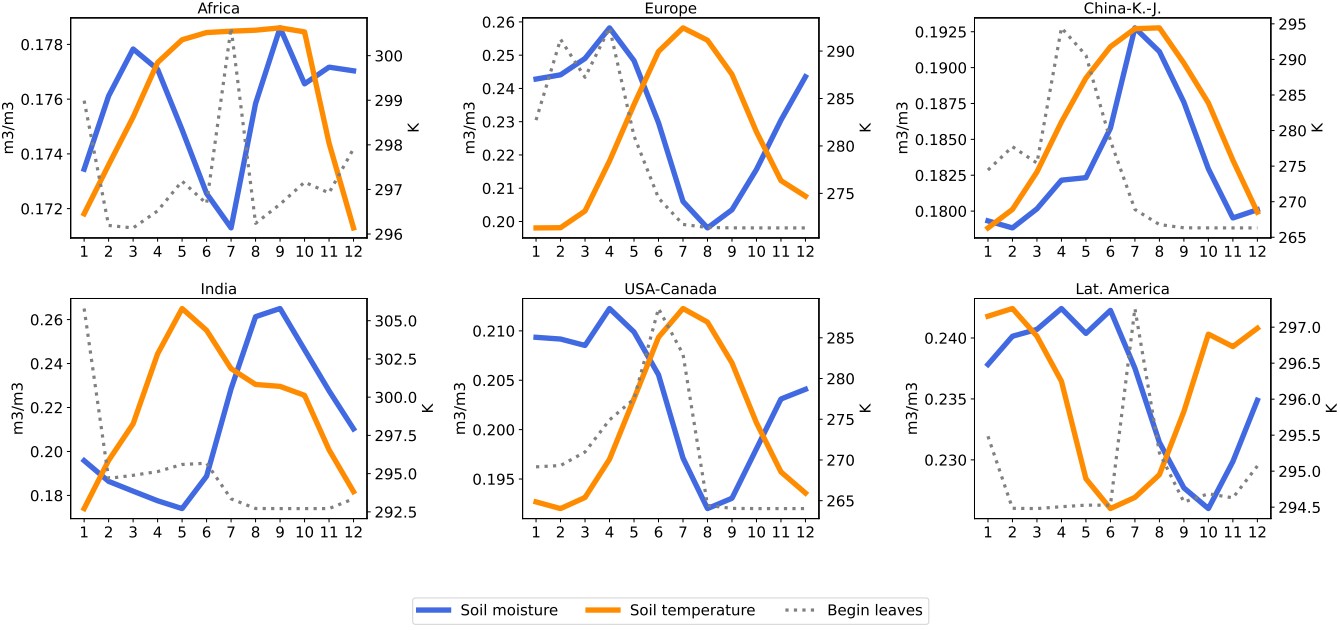

**Figure 10.** Regional monthly simulated soil moisture ($m^3/m^3$) in blue and soil temperature (K) in orange. Regional monthly signal for leaves to start to grow in ORCHIDEE averaged over the different PFTs is drawn with dotted grey lines (this metric has no unit and can be seen as a qualitative signal for the start of the vegetation growth season). Variables are for 2006. China-K-J accounts for China-Korea-Japan.

Last, higher emission factor values imply an increase of total $NH_3$ emissions of about 6% globally. Although this change is not as significant as other factors, it is worth noting that this impact is only driven by indoor emissions, which account for 22% of the total emissions globally. In Europe and Latin America, for instance, where the contribution of indoor emissions to the total emissions can reach more than 90% (Fig S6. in the Supplement), the impact of using higher $EF_{max}$ values calculated at the scale of these two regions (14% and 12%, respectively) is higher than the impact calculated at the global-scale.

## 3.5 Emission seasonality

Seasonality patterns have been first explored by comparing our emissions against the CEDS inventory and the FANv2 simulated emissions. As shown in Fig 11, we calculate maximum emissions during the spring and summer seasons, while in CEDS and FAN, the emissions peak almost everywhere only during spring. In the three datasets, the lowest values are calculated during winter, when the meteorological conditions are not favorable to the emissions, and the N application is the smallest.

The summer peak observed in our emissions and the spring peak in CEDS and FAN in the USA, and more specifically in the central and south central parts, are also reported by the MASAGE bottom-up inventory from Paulot et al. (2014). In MASAGE, 2 peaks are highlighted during the year, one in March and the other in June. Goebes et al. (2003) and Pinder et al. (2006) attributed these peaks to the timing of the mineral fertilizers and manure application. It is consistent with our approach since indoor emissions do not vary over the year, and only the N applications are time-dependent. In Europe, our emissions are higher





in summer while Paulot et al. (2014) estimate a clear peak in spring, like in FANv2 and CEDS. In addition, the analysis of Fortems-Cheiney et al. (2020) based on different emission inventories for France shows the substantial contribution of mineral fertilizer application on emissions leading to a peak in April. Paulot et al. (2014) demonstrates that April is when emissions

are maximum over several European agricultural regions such as Portugal and Spain or Benelux, Germany, and Denmark due to local regulations preventing farmers from applying manure outside the growing season. These regions are also characterized by large emissions in July, likely coming from livestock. It has been recently confirmed by top-down emission based on CrIS and IASI observation estimates in the UK (Marais et al. (2021)). Our approach is constrained by the low level of detail in crop diversity, mainly due to the spatial resolution of the model. Thus, we choose a long enough N application period to catch the

crop system diversity. In China, the highest emissions are calculated by our model in summer, which is supported by previous inventories (Streets et al., 2013; Kang et al., 2016; Xu et al., 2018) and satellite observations such as the TES instrument (Shephard et al., 2011) and the AIRS retrievals (Warner et al., 2017). Our emissions in India peak in spring but remain high during summer. This pattern is also highlighted in the HTAP emissions and IASI satellite data (Janssens-Maenhout et al., 2015; Van Damme et al., 2017) where there is no strong seasonality shown over the Indo Gangetic Plain but higher emissions from

April to September.

To complete our analysis, monthly emissions derived from the IASI satellite ($IASI^{inv}$) averaged over the 2011-2015 period are used as a comparison regarding different hot-spot regions defined in Fig 5. The same operation has been done to agricultural emissions from FANv2 and CEDS inventory and the detail of the $CAMEO_+$, $CEDS_+$ and $FAN_+$ dataset constituents is listed in Table 6. First, it is worth noticing that the seasonality of $CAMEO_+$, $CEDS_+$ and $FAN_+$ are primarily due to their

respective agricultural emissions; the other sources have no important role in the seasonality. Indeed industrial and waste sources show very low annual standard deviations (in the ranges of $10^{-7}$ and $10^{-8} gNm^{-2}yr^{-1}$ respectively) compared to the agricultural emissions estimated by CEDS, FANv2 or CAMEO ($0$-$8gNm^{-2}yr^{-1}$). Biomass burning emissions have a standard deviation reaching $0.6gNm^{-2}yr^{-1}$ in areas characterized by a high fire activity while in agricultural regions, the deviation is intermediate ($\sim10^{-2}gNm^{-2}yr^{-1}$, Fig S10. in the Supplementary Material). Overall, our emissions seem to be consistent with

$IASI^{inv}$, with the general patterns and the absolute values being similar (Fig 12), except for Eq. Africa and Mid-Brasil regions where $CAMEO_+$ largely underestimate $IASI^{inv}$. Emission seasonality patterns in $CEDS_+$ and $FAN_+$ are quite close from each other, but very different from $IASI^{inv}$. $FAN_+$ and $CEDS_+$ usually depict a sharp peak in spring (one month in advance in $FAN_+$), and another smaller peak only for $CEDS_+$ in September-October. While the seasonality in $CEDS_+$ is artificially retrieved from a specific profile, the temporal variability in $FAN_+$ is driven by the meteorological conditions and the crop

types present within the pixel. Even though the representation of crops in the CLM5 used in FANv2 is more precise than in ORCHIDEE (8 crop types against 2), we observe that the temporal variability is better represented in our approach when compared to $IASI^{inv}$. In FANv2, the fertilizer application is triggered 20 days following the leaf emergence. It might explain the single sharp peak observed in spring, while the long period (6 months) we have chosen seems to capture the general pattern of emissions better. More specifically, there is a very good agreement between $CAMEO_+$ and $IASI^{inv}$ in the Mid-USA where

emissions peak in summer ($>2gNm^{-2}yr^{-1}$). In Europe, our calculated emissions show a clear and strong peak in July, while $IASI^{inv}$ patterns are different. In Northern Europe, $IASI^{inv}$ present two peaks in March and August with very low emissions





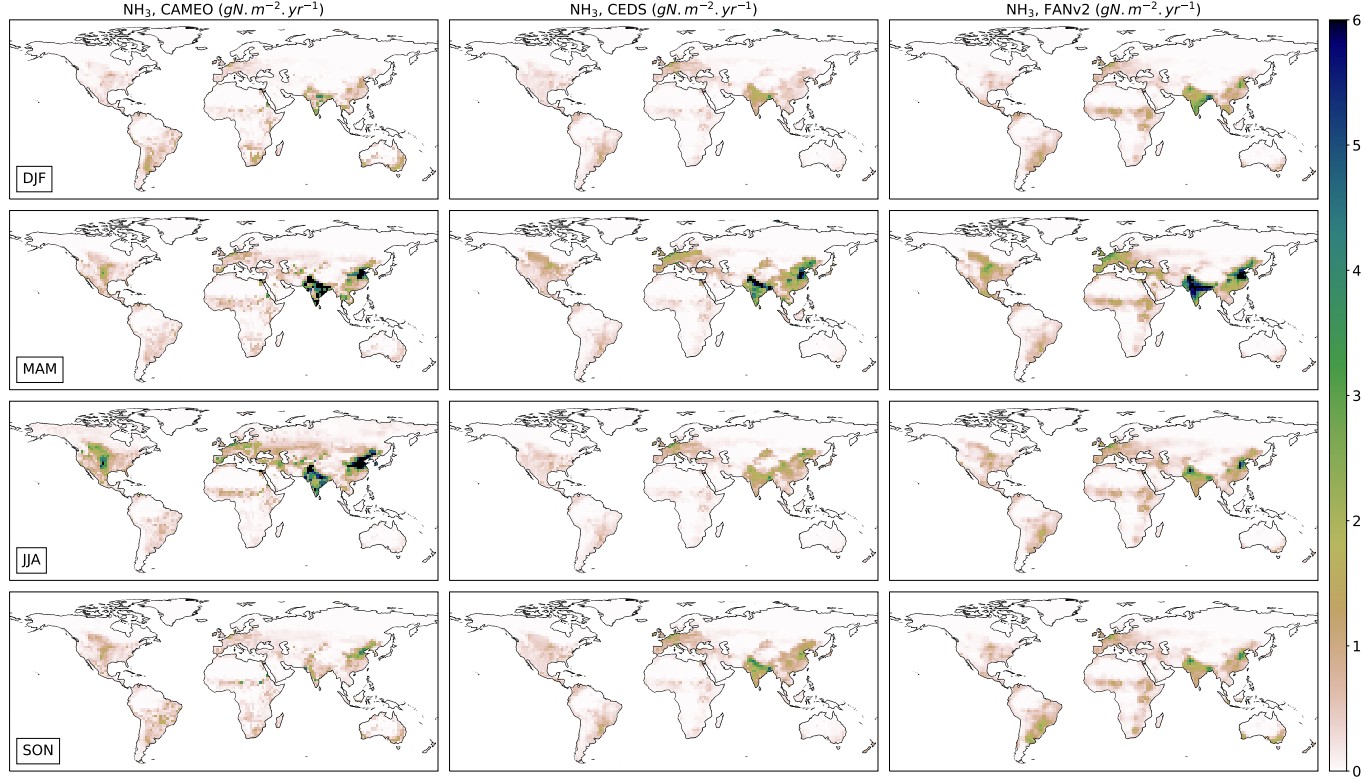

**Figure 11.** Seasonal patterns of ammonia emissions ($gNm^{-2}yr^{-1}$) from total agricultural sources simulated by CAMEO (2005-2015 average, first column), estimated in the CEDS inventory (2005-2015 average, second column) and the FANv2 model (2010-2015 average, third column).

in winter while in $CEDS_+$, $CAMEO_+$ and $FAN_+$, emissions are maintained around 0.5-0.8$gNm^{-2}yr^{-1}$. These differences might be explained by high uncertainties associated with the IASI instrument during this period. Van Damme et al. (2014) highlights limitations in the IASI (MetopA) measurement availability over Europe in winter 2011. The number of cloud-free

observations is low, especially in December and January, with only 4% of the dataset being associated with an error lower than 50% due to the thermal contrast (defined as the temperature difference between the Earth's surface and the atmosphere at 1.5 km) and the amount of ammonia present in the atmosphere. In Southern Europe (Spanish region), $IASI^{inv}$ emissions show a weaker seasonal cycle than $CAMEO_+$ peaking in summer. Spain is characterized by a diversity of agricultural production types (crops, fruits, olives, etc.) which are not differentiated in our model. It shows a limitation in our approach where the

representation of agricultural lands seem homogeneous and induce that the resulted emissions have the same seasonal pattern over this region. In the northern part of India, $CAMEO_+$ highlights a peak in May, which is two months earlier than the one in the $IASI^{inv}$. In agreement with $IASI^{inv}$ results, Tanvir et al. (2019) also depicted a clear peak in $NH_3$ concentration in July by using the TES data in the Indo-Gangetic-Plain. It is worth noticing that IASI observations in this specific region might be



associated with a high level of uncertainties (Marais et al., 2021). Marais et al. (2021) indicate that in addition to the intense
biomass burning season and the relatively low abundance of acidic aerosols in northern India, warm temperatures may increase
the emissions and suppress partitioning of $NH_3$ to aerosols inducing an enhancement in the spectral signal. In the Chinese
hotspot, we observe the same peak in summer for both $CAMEO_+$ and $IASI^{inv}$ even though our maximum value is almost 2
times higher than the one in $IASI^{inv}$ (Fig 12).

In Africa and Latin America, our emissions show a less pronounced variability over the year than $IASI^{inv}$. Results in
Africa show 3 peaks (February, May, and October) in the $IASI^{inv}$ while our emissions highlight only one peak in June (Fig
12). Equatorial Africa is a specific region in emission seasonality that has been recently studied in Hickman et al. (2021).
They reveal different ecoregion drivers of the atmospheric $NH_3$ explaining seasonal patterns observed by the IASI satellite.
The region chosen in our study is between the two northern ecoregions: wet and dry savannas, which are characterized by
important livestock densities and intense biomass burning activities.

The June peak retrieved in $CAMEO_+$ and $IASI^{inv}$ can be attributed to the seasonal pattern of the dry savanna. This time
of the year corresponds to the rainy season in the dry ecoregions, and emissions from soil (from livestock excreta and natural
processes) are expected to be stimulated through the microbial activity enhanced by precipitations (Hickman et al., 2018).
Hickman et al. (2021) demonstrated that precipitations and temperature are the most important predictors to explain the sea-
sonality of the emissions in this region. The two other peaks in the $IASI^{inv}$ during the dry season (February and October) can
be a contribution of the wetter region located just below the dry savanna. Hickman et al. (2021) demonstrated, in addition to
the importance of the soil emissions, in this region, vegetation fires during this period might explain additional emissions. This
result is also supported by the long-term measurements from the INDAAF network, where it is suggested that the seasonality
in the wet savanna is the result of biomass burning with a high increase in the concentrations during the dry season (Adon
et al., 2010). The absence of these two peaks in our estimate can be explained by an underestimation of the biomass burning
emissions from the GFEDs4 inventory used to complement our emissions (van der Werf et al., 2017). This inventory is based
on MODIS biomass burning area, and recent analysis suggests that MODIS underestimates fire emissions by a factor of 2-5
because of the non-detection of small fires (Roteta et al., 2019; Hickman et al., 2021; Ramo et al., 2021) mainly coming from
agricultural practices.

The case of Latin America has been much less studied, and we observe strong seasonality in the $IASI^{inv}$. The two regions
in Latin America are characterized by croplands, intensive livestock farming, and biomass burning activities. Over the Mid-
Brasil, $IASI^{inv}$ reveal an important peak in September ($>4gNm^{-2}yr^{-1}$) which is only represented in $CAMEO_+$ and $FAN_+$
time-series by a smooth increase. Andela et al. (2017) has shown a strong positive spatial correlation between burned area
and cropland fractions in this ecoregion, probably suggesting significant agricultural waste burning. In addition, **?** also shows
a maximal biomass burning activity measured by MODIS via the monthly mean number of fires (MODIS fire dataset) in the
Brazilian Caatinga shrub lands and in the north-eastern part of the Cerrado region in September. In the "Pampas" region, the
general seasonality from the $IASI^{inv}$ and our emissions describes the highest emissions from September to March and low
emissions the rest of the year. However, our emissions do not highlight the clear peaks in March and September revealed in
the $IASI^{inv}$. The work of **?** points out the sugarcane and soybean expansions in this region as the main drivers of biomass




burning. It can lead to the same conclusion as in Africa concerning the small fires, which are hardly detected in MODIS and

might be not considered in the GEFDs4 inventory used in our work. It is also worth noticing that in Argentina, manure applied as fertilizer is not a common practice (Vázquez Amabile et al., 2012). In our approach, all the manure is applied. It would potentially lead to a distinct seasonal cycle compared to a case where all the manure is stored the whole year.

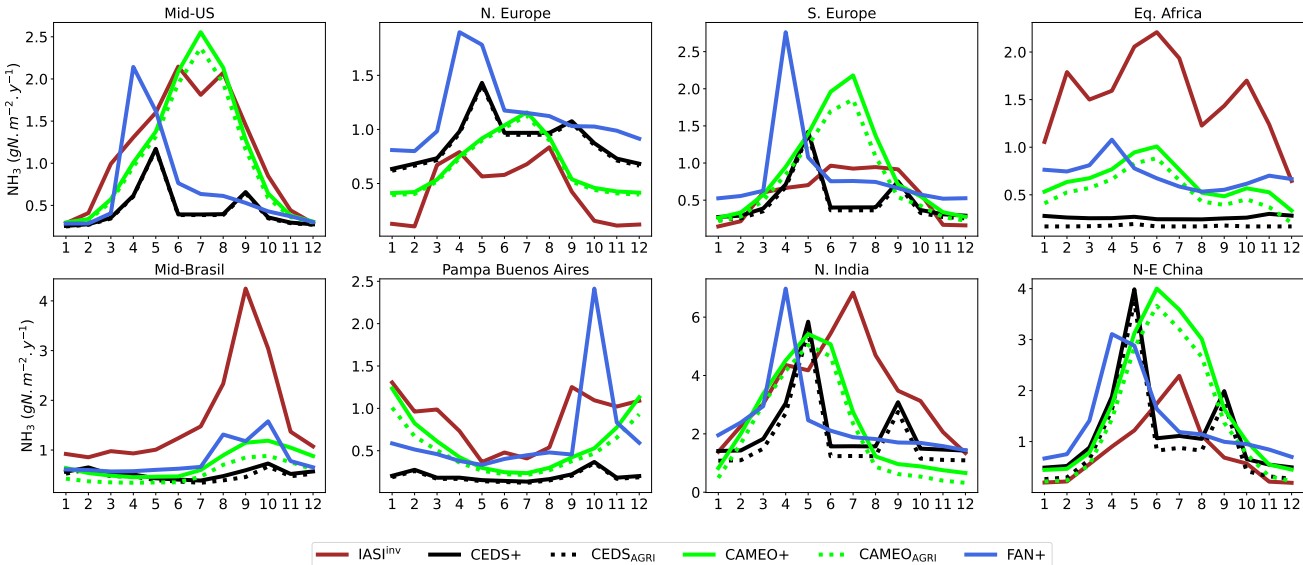

**Figure 12.** Monthly regional $NH_3$ emissions ($gNm^{-2}yr^{-1}$). CAMEO emissions accounting for natural and agricultural emissions aggregated with other sources is represented by the solid green line ($CAMEO_+$) while agricultural emissions by the dotted green line ($CAMEO_{agri}$). The agricultural sector of CEDS alone and aggregated with other sources are represented by lines in black dotted ($CEDS_{agri}$) and solid lines ($CEDS_+$) respectively. The $IASI^{inv}$ product is in red ($IASI^{inv}$). The agricultural emissions from FANv2 aggregated with other sources are shown in blue ($FAN_+$). Other sources include biomass burning from van der Werf et al. (2010) and industrial and waste sectors from CEDS. Regions are defined in Fig 5

. .

The temporal correlation scores between the reference (here : $IASI^{inv}$) and $CAMEO_+$ , $CEDS_+$ and $FAN_+$ calculated over the monthly time-series on the 2011-2015 period are plotted in Fig 13. Results highlight excellent month-to-month variability

agreement between $CAMEO_+$ and $IASI^{inv}$, in most regions of the globe. In the main hot-spot regions, such as China, India, Europe and the USA, correlations are comprised between 0.7 and 0.9 while correlations between $IASI^{inv}$ and $CEDS_+$ and between $IASI^{inv}$ and $FAN_+$ hardly exceed 0.5. It means that our modeling approach enables a satisfying representation of the seasonal cycle in terms of agricultural and natural emissions in comparison with CEDS agricultural emissions, where a forced seasonal profile (2 high peaks of volatilizations in May and September) is used and with the FANv2 model, accounting for a





more realistic representation. However, in the south-eastern part of the USA and the Chacò region in Latin America, we observe
a degradation of the seasonal pattern in our emissions compared to both $CEDS_+$ and $FAN_+$ where the correlations are high.
The Chacò region is one of the main hot-spot in terms of natural soil emissions in our model, as highlighted in Fig S8. from the
Supplementary Material. It is characterized by savanna with grasslands, thorn forests, a mosaic of woods with savanna, shrubs,
and coarse grass predominate (Berry et al., 1995). However, most of the natural emissions computed in ORCHIDEE originate

from temperate broad-leaved summer green PFT. Many studies demonstrate the importance of the biomass burning events in
the emission quantities mainly occurring during the dry season in September - October (**?**Pereira et al., 2022). $IASI^{inv}$ depict
an important peak in the $NH_3$ emissions (See Fig S9. in the Supplementary Material) during this time of the year and can
be attributed to fire events. We observe that natural and agricultural emissions have very similar patterns and are in the same
range. However, fire contribution in $CEDS_+$ and $CAMEO_+$ datasets appears to be very low, supporting the limitation of using

the GFEDs4 inventory in bottom-up $NH_3$ emission estimates to be compared with $IASI^{inv}$. The degradation of the correlation
between $IASI^{inv}$ and $CAMEO_+$ compared to the one with $CEDS_+$ in the Chacò region is explained by the fact that there is
almost no temporal variability in the $CEDS_+$ at year scale.

In India, there is an interesting pattern with a clear longitudinal delimitation with western negative and eastern positive
correlations in $CAMEO_+$ and $CEDS_+$. In the northern-western part of India, $CAMEO_+$ performs better at capturing the

$IASI^{inv}$ seasonality than $CEDS_+$ and $FAN_+$.

Based on the comparison with the seasonality of the $IASI^{inv}$ there is a strong limitation in using CEDS as $NH_3$ emission
information for CTM in order to study its impact on the atmospheric chemistry. More specifically, we demonstrate that using
ORCHIDEE land-based emissions has the potential for improving the seasonal signal of the resulted ammonia concentration
in the atmosphere.

However, using $IASI^{inv}$ to evaluate our model results also have limitations, due to the uncertainties associated with the
satellite product and the derivation method. For example, many studies using IASI data do not consider winter observations
in the USA and Europe (Marais et al., 2021), due to a potential degradation of the data because of atmospheric conditions
(cloud coverage, low temperature etc). The use of the $NH_3$ lifetime simulated by LMDZ-INCA in the inversion method is also
associated to an uncertainty. Finally, the interpolation method used to regrid the IASI observations onto the LMDZ-INCA grid

can also be a source of uncertainties as demonstrated in Evangeliou et al. (2020).

## 4    Conclusions

In this study, we implement a new module dedicated to global $NH_3$ emissions from agricultural practices including livestock
waste management and mineral fertilizer application within the ORCHIDEE land surface model. Our development allowed
to consider dynamical variables (such as surface temperature and humidity) through different physical soil processes for the

calculation of the $NH_3$ fluxes. This aspect, often neglected in bottom-up approaches, is key for a realistic seasonal represen-
tation of the emissions. In contrast to other emission models, our module interacts with the ORCHIDEE model for vegetation
variables such as the biomass productivity. It allows the calculation of a grazing index, a global indicator of the pressure exerted



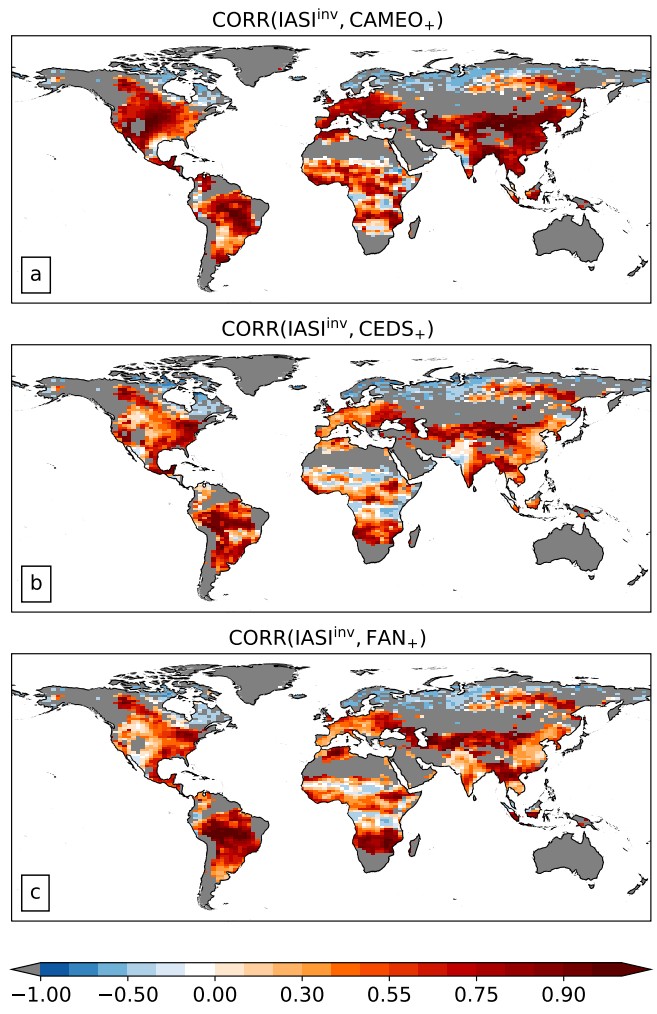

**Figure 13.** Temporal correlation scores between IASI$^{inv}$ and CAMEO emissions aggregated with other sources (a) and between IASI$^{inv}$ and total emissions from CEDS for agricultural, industrial and waste sources aggregated with biomass burning (b) over 2011-2015. Temporal correlation scores between IASI$^{inv}$ and FANv2 aggregated with other sources are shown in (c). Other sources include biomass burning from van der Werf et al. (2010) and industrial and waste sectors from CEDS. Grey grid-cells correspond to a standard deviation in the monthly IASI$^{inv}$ time-series lower to $0.2 \mathrm{gNm^{-2}yr^{-1}}$ in order to avoid biased correlation scores.

on the vegetation. We estimate global agricultural emissions of about $44 \mathrm{TgNyr^{-1}}$ with soil volatilization from fertilizer and manure applications accounting for $78\%$ and indoor emissions (from livestock housing, yard and storage) for the remaining.
The spatial distribution of the calculated emissions is consistent with previous studies (the bottom-up inventory CEDS (Hoesly et al., 2018) and the process-based model by Vira et al. (2019)) highlighting the most important emitting regions such as Eastern China, Northern India, the USA and Europe characterized either by high N application rate or intensive livestock farming.



In order to evaluate the modeled emissions, different sensitivity simulations involving key parameter variations have been performed. The most important parameter driving the emissions is the pH of the N input, which induces an increase of about
74% of NH$_3$ emissions when shifted from 7 to 7.5. Assuming a constant value for the pH simplifies our approach. Using a soil pH map would imply more complex processes involving a change in the pH during N application. In addition to the ammonium content, the pH and the type of fertilizer used are hardly available in the literature. Manure management emissions are also associated with uncertainties from the use of EFs. Even though EFs are calculated with an extensive definition of livestock and management systems, and considering variations in climate and management practices, it is an important assumption
to use European EFs for the whole globe. However, we demonstrate that the overall emissions are moderately sensitive to the EFs, with a global change of 6% of the emissions when the maximum range given by Sommer et al. (2019) is used. Similarly, regional parameters given by FAO (2018) were simplified to match the representation of the vegetation distribution in ORCHIDEE. Assuming livestock feeding composed of only grass and crop products, neglects the use of the agro-industrial by-product, which is standard practice in Europe and the USA. Modeled emissions are also sensitive to the timing of the N
application, especially in China, where a shortening of the fertilization period induces very high volatilization rates. Apart from this parameter, none of the other factors tested appear to be important drivers of the emission seasonality. Finally, the seasonality patterns have been further analyzed by using satellite derived emissions. The comparison suggests that ORCHIDEE simulates a very good representation of the seasonality of NH$_3$ emissions, with correlation scores larger than 0.7 in the most important emitting regions.

In addition to the gain in realistic seasonality, our approach fills the lack of estimates for emissions from natural soils missing in almost every inventory. It is highly interesting for Africa and Latin America, where these sources are important (Hickman et al., 2018) and little studied. These encouraging results prove the potential of coupling ORCHIDEE land-based emissions to CTMs, which are currently forced by bottom-up anthropogenic-centered inventories such as CEDS. This framework allows room for improving the representation of the emissions since atmospheric variables are dynamically simulated by CTMs. For
instance, the surface NH$_3$ concentrations used in the final calculation of the emissions could be updated at each time step instead of prescribing an external monthly file for a given climatology. In addition, it exists a tight relationship between emission and deposition of NH$_3$ since NH$_3$ is particularly reactive and deposition of NH$_4^+$ contribute to the re-emission of NH$_3$ from natural and managed soils. By coupling the emissions with the global CTM LMDZ-INCA through the dynamic calculation of wet and dry depositions, we plan to improve the representation of the emissions as well as the atmospheric concentrations. In addition,
a further evaluation will consist of comparisons with atmospheric composition observations.

*Code and data availability.* The ORCHIDEE model is available at https://forge.ipsl.jussieu.fr. The modified version of ORCHIDEE including the CAMEO module used in this paper is available at https://forge.ipsl.jussieu.fr/orchidee/wiki/GroupActivities/CodeAvalaibilityPublication/ ORCHIDEE_CAMEO_gmd_2022 (Beaudor et al., 2022). The NH$_3$ emissions simulated by CAMEO and the manure produced along with soil ammonium concentrations are available at https://doi.org/10.5281/zenodo.6818373; other model outputs and the IASI derived emissions



are available upon a reasonable request from the authors. The NH$_3$ emission inventories used in this study are available in Hoesly et al. (2017) for the CEDS and https://doi.org/10.5281/zenodo.3841723 for the FANv2 data.

*Author contributions.* NV, DH, JL and MB designed the module. NV and MB implemented the module in ORCHIDEE. NV and MB designed and performed the simulation experiments. MB analyzed the output and prepared the manuscript with contributions from NV, DH and JL. MVD and LC provided the IASI satellite product and performed the regridding of the data. NE conducted the emission derivation
from the IASI product. All the authors contributed to the writing of the manuscript.

*Competing interests.* The authors declare that they have no conflict of interest.

*Acknowledgements.* We acknowledge the support by the ESM2025 project. This project has received funding from the European Union's Horizon 2020 research and innovation programme under grant agreement N° 101003536. We also acknowledge the support by the supercomputer system of GENCI (Joliot Curie supercomputer). N. Evangeliou was supported by the Research Council of Norway (project ID: 275407,
COMBAT – Quantification of Global Ammonia Sources constrained by a Bayesian Inversion Technique). M. Van Damme is supported by the FED-tWIN project ARENBERG ("Assessing the Reactive Nitrogen Budget and Emissions at Regional and Global Scales") funded via the Belgian Science Policy Office (BELSPO). L. Clarisse is Research Associate supported by the Belgian F.R.S.-FNRS.





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
