# Peer review of "Global agricultural ammonia emissions simulated with the ORCHIDEE land surface model"

_EGUsphere, 2022_

## Author Response (AR1)

We thank both reviewers for their comments on our manuscript and for the time spent on their reviews. Please find below a detailed point-by-point reply to the comments and suggestions to reviewer #1 and reviewer #2.

In the following, reviewers' comments are in black, whilst our responses are in green. The text added in the revised version of our manuscript is in italics and the line numbers correspond to the first version of the manuscript.

**Reviewer1**

Specific comments:

☐ Modeling set-up: I think it would be very helpful to have a more detailed description of the model run set-up in the supplementary showing how the tested parameters are integrated in the model.
We agree it is important to indicate where the parameters are integrated into the model, and for that, we suggest to add a column in Table 5.
In this additional column, we precise the equations and the sections in which the parameters are tested.
When the parameters are not involved in any equations, we rather add a description of the parameter in the main text, and the parameters are referenced by its corresponding Section in Table 5.
This sentence has also been added for clarification:
L.266 *It is worth noticing that $input_{fert}$ is given as the ammonium content of the total N mineral fertilizer applied (the parameter $Frac_{nh4,fert}$ is the fraction of ammonium content of the N fertilizer used to make the conversion, this parameter is tested in the sensitivity analysis).*

☐ L672: fertilizer types are available from IFASTAT so I am not sure this statement is true
"In addition to the ammonium content, the pH and the type of fertilizer used are hardly available in the literature."
We agree fertilizer types are available. We suggest another formulation:
*In addition to the ammonium content, the pH used is hardly available in the literature.*

☐ L328: sensitivity not sensibility done
☐ Table 5: time steps : done
☐ L354: TRENDY (Le Quere et al., 2018) : done
☐ L347: sheep done
☐ L380: Have you compared total C production? Maybe showing this comparison would be helpful when arguing that the C:N ratio is the main difference. And could there also be an issue with legumes in grassland that you cannot represent in ORCHIDEE?
We thank the reviewer for this interesting comment regarding the grassland C productivity estimated by the model.
This data is indeed hardly available in the literature; none of the references used to evaluate our intermediate variables report estimates of C content. They mostly focus on N quantification.

However, Figure 11.9 from the AR5 report of the IPCC (Climate Change; Smith et al., 2014) mentions total grass biomass grazed/harvested of around 1.95 PgC $yr^{-1}$ (after a conversion from dry matter to C assuming 0.5 as a conversion factor).
In our approach, we estimated global grazed biomass of around 1.2 PgC $yr^{-1}$ which also suggests a significant underestimation of the grass productivity in C (~40%).
Nevertheless, a C:N ratio model overestimation is also a plausible reason for underestimating N grass biomass productivity since we have shown, based on the literature, a much stronger underestimation in the case of N biomass productivity, compared to the C productivity
It is also worth noticing that the N content in grassland vegetation estimated in previous studies such as in Billen et al., 2014, can present some uncertainties.
Billen's global estimate (80 TgN $yr^{-1}$ ) is in fact based on an indirect calculation resulting from a simple difference between livestock ingestion and available crop feed resources. Therefore, it is likely that N grass biomass is underestimated in our model,  due to an overall underestimation of biomass productivity (C and N), and possibly to an overestimation of the C:N ratio. However, it is hard to quantify what can be the contribution of the C:N ratio in the overall N biomass underestimation. As the reviewer suggested, a better representation of the legumes for instance, could help in providing more N through the BNF into the ecosystems and reaching lower C:N ratios.

In addition, we are aware of some regional gaps where the grassland area is low in ORCHIDEE, as in India, which may also lower our global model estimate of the N grass biomass production
We suggest to add these 2 sentences:

*L378: By doing so, uncertainties from several components (crop production, net import of vegetal proteins, and human consumption of vegetal proteins) are accumulated.*
*L381: However, if the grass N production is largely underestimated by ORCHIDEE, our grass C production estimate of 1.2 PgC $yr^{-1}$ is close to the value of 1.95 PgC $yr^{-1}$ reported in the  IPCC AR5 report (Climate Change; Smith et al., 2014).  In this respect, an overestimation of the C:N ratio may also explain part of the grass N production underestimation.*

☐ L381: I do not quite understand how your excretion rate can be smaller than the values given by Paustian et al. (2006). Looking at (5), it seems like you took the excretion rate from Paustian et al. (2006).
It is true we compared a regional indicator as an excreted biomass per 1000 kg of animal (from Paustian et al., 2006) and we also took as input for the model an excretion rate expressed as a percentage of the ingested biomass (both from Paustian et al., 2006).
We agree that our comparison of the excretion rates can be confusing for the reader and since only regional information is provided in Paustian et al., 2006 we prefer removing this comparison point from the section where only global budgets of N are presented. Thus we remove these 2 sentences in the main text (L377 and 381). In addition, the

lines corresponding to ER in Table 6 and 8 have been removed.

☐ L385: As far as I understand and as I can see in Table 8, what you describe as manure production (66Tg) is manure application. If this is the case, I would not compare it to global estimates of manure N excretion but rather to estimates of N application as well. If this is really manure application, I would also rephrase this sentence: L384: 'In our calculation the manure produced is directly applied to soil'
We agree with the reviewer; we should only consider from the literature the manure which is applied to the soil since in our approach we do not consider any other pathway for the final stored manure.
We suggest the following reformulation for sentence L384:
*Because we assume that all of the manure stored is then applied to soil, we only consider for the evaluation phase literature data which estimates manure application rate.*
In addition, Table 8 has been modified in order to keep only the variable corresponding to the applied manure.

☐ L431: agricultural NH3 emissions done
☐ L432: half instead of twice lower done
☐ L478: soil pH instead of just pH might be better for clarification. We corrected it by mentioning 'manure pH'.
☐ There are question marks where references are supposed to be throughout the paper. Please check and add the respective references. done

**Reviewer2**

Comments

Areas identified for major improvements in order to be accepted for publication.

● Indoor ammonia emissions. Units for equations in this section (pages 8 and 10) need to be clear, particularly TAN related. Emission factors in Table 3 (many > 1) as factor of TAN and it is hard to understand why they are great than 1 from the units provided. We thank the reviewer for this clarification. Indeed, these units need to be corrected in the table caption (done). In Table 3, EF are presented as % of TAN content in the manure. However, in the equations, we converted the EF as kgN/kgTAN which corresponds to a proportion rather than a percentage.

● Soil ammonia emissions

1. It is hard to understand the $Z_{activity}$ parameter in equation 13 as it is on both side of the equation. How is TAN(soil,aq) related to this parameter? I would think deposition is surface application like fertilization despite that some fertilization is applied in deep soil to avoid surface runoff.
We agree with the reviewer, we should rather express the $Z_{activity}$ as a function of the time since it is updated at each time step.

It is now corrected in the manuscript as followed: $Z_{activity}(t) = X * Z_{activity}(t-1)$ .

The Zactivity parameter is more related to the N concentration in the liquid phase than strictly to the depth level at which N is applied. We fully agree deposition should be considered as a surface application. By default, we assume that the activity layer for N dynamic equals 0.2m. This assumes that all the TAN is located within this layer at a given concentration depending on the soil water content. When applying fertilizer and manure, we have information on their specific N concentration that we directly use to set the pzact_surf variable. For NHx deposition, currently, we have no data regarding the specific water volume of NHx deposition and no information either on how to treat dry and wet deposition. As a consequence, we prefer not to specify any particular parametrization regarding its concentration and instead use the default parametrization.

2. The ammonia flux equation (16) is bi-directional depending on the free-atmosphere concentration which changes seasonally and diurnally. It is too crude to use monthly field averaged over 11 years from the global run (LMDZ-INCA) for its 30min simulation (acknowledged in the conclusion). Although the sensitivity test on this field did not show significant change comparing the change in pH and days of fertilization probably due to averaging evaluation, it does not mean it is not important. Since this is a key parameter in flux calculation, more evaluation is needed. For instance, the paper needs to address how it treats negative and positive flux (16) (average or only count positive flux as emissions). How good is the free-atmosphere concentration – any evaluation comparing ammonia flux field measurement? Or, maybe one year simulation with the free-atmosphere concentration directly from the global run (not averaged) should be conducted to evaluate how it influences the soil emissions spatially and temporally.

We know the uncertainties related to the fixed atmospheric concentration in the emissions calculation.

In the paper's framework, ammonia emissions are estimated from an offline point of view through a surface model.

In our current approach, N depositions are considered through the soil TAN pool, which is involved in the calculation of the gaseous $NH_3$ concentration.

However, no proper compensation point is implemented yet and only two resistances are represented (aerodynamical and quasi-boundary layer resistances).

Since no coupling between the atmosphere and surface is not yet fully implemented around the N cycle, forcing ORCHIDEE by a fixed concentration is the most obvious option, as no interaction with the atmosphere exist. The monthly time-step has been chosen due to computational constraints. As a global land surface model, ORCHIDEE commonly receives monthly or annual forcing files (N fertilization, N deposition, BNF, $CO_2$ concentrations), except for the meteorological fields where a pre-processing work has been performed to adapt the data to the model.

We are currently working on the coupling between LMDz-INCA and ORCHIDEE, both components of the IPSL ESM.

In addition to the transmission of the hourly-calculated fields as N depositions and $NH_3$ concentrations from the atmosphere to the surface, surface compensation points will be implemented to integrate more accurate bi-directional exchanges of $NH_3$.

Therefore, the influence of several key variables will be tested and compared against the

offline mode.

- Constant pH. Giving the importance of pH in soil ammonia flux modeling – demonstrated in many publication (e.g. Pleim et al., 2019, JAMES), it seems that there is no reason to use a fixed pH in this global-scale modeling. Using the soil pH map directly would be a better sensitivity test than just changing it to another constant higher (7 to 7.5) – clearly high emissions expected.
  We thank the reviewer for this remark. We are aware of the simplification done by taking a fixed pH to calculate the $NH_3$ volatilization.
  Our understanding of the influence of the pH is linked to the modification of the soil pH after the application of N input and thus through the pH of the manure or fertilizer, as indicated in Massad et al., (2010).
  In their paper, they also mention that depending on the fertilizer type, the pH of the solution might not be impacted by the soil pH (e.g, ammonium nitrate).
  Despite the fact that ORCHIDEE is forced by annual soil pH maps, there is no update of the soil pH related to N input, and the soil pH can be much lower than in reality.
  To be more realistic we should consider the perturbations in pH since the N addition passes, and their magnitudes depend on the type of manure or fertilizer as described in Vira et al., (2019).
  However, this implementation is complex and is not part of the N cycle that we aimed to improve in this paper.

- Figure maps are too small and have color scales difficult to see the regional differences (figures 2, 4, 5, 6, 8, 11, 13 and those in the supplement). done

More specific minor comments are listed below:

- Spell out acronyms in the abstract (e.g., ORCHIDEE, USA, CTM, CEDS). done
- Spell out acronyms in the main text (e.g., CEDS and EDGAR in line 59, FAN in line 74…, and many others). done
- Not all grassland is for grazing or hay production. How does the system differentiate grassland in the grid cell to be natural grassland or for agricultural production? This is related to whether all grassland in the grid cell receive fertilization – both manure and synthetic.
  ORCHIDEE does not differentiate natural from managed grassland.
  It is exact, only grid cells either with the presence of livestock or fertilizer application are considered for the emission calculation, so we can assume that the pixel is managed.
  Following sentence is added L.183:
  *Please note that ORCHIDEE does not differentiate natural from managed grassland. Only grid cells either with the presence of livestock or fertilizer application are considered for the emission calculation, so we can assume that the pixel is managed.*

- How does the system constrain each grid-cell's effective crop biomass by the global crop harvested NPP – explain more (lines 175-176)?
  We compute the ratio between the global effective crop biomass and the global crop harvested NPP (HI) at a yearly time-step. When HI >1, we impose the same constraint to the global effective crop biomass at each grid cell by dividing to HI. However, this condition never occurs in this simulation and it is rather considered as an indicator, especially for future scenarios where the biomass simulated by ORCHIDEE can be a constraint to support future livestock.
  The following sentences has been added to lines 177-178 :
  *To do so, we compute the ratio between the global effective crop biomass and the global crop harvested NPP (HI) at a yearly time-step.*
  *When HI>1, we impose locally the same constraint by dividing the effective crop biomass by HI.*

- N by plant uptake in the agricultural land is the biggest out pathway for N leaving the field (e.g., Ran et al., 2019, JAMES). The paper needs to address how N uptake is handled for fertilized cropland and grassland.
  We thank the reviewer for suggesting information about plant N uptake. Plant N uptake is modeled based on the work of Zaehle et Friend (2010). Plant N uptake is modeled as a function of N available in soil but also of root biomass. The more N in soils or the more root biomass, the higher the plant N uptake. The plant N uptake modeling accounts for also information regarding the plant N status leading to higher N uptake for N starvation conditions.
  Following sentence is added line 110 of the first version of the manuscript:
  *TAN pool is also updated according to plant uptake as described in* Zaehle et Friend (2010)

- N fixation is associated with specific grassland (e.g. alfalfa) and cropland (e.g. soybean). Does the data used in the system target the N fixation grassland or cropland (lines 343-344)?
  Thanks for the comment. Indeed, in our modeling framework, BNF is only considered for natural ecosystems, not for managed ones. Consequently, BNF implied in leguminous systems such as alfalfa or soybean are not considered. This potentially may limit plant productivity for regions with no use of synthetic fertilizers and where leguminous species are important.
- Many question marks in the text (e.g., lines 415, 417, 613, 618…) – correct them.
  done

---

## Author Response (AR2)

We thank the reviewer for his comments on our manuscript and for the time spent on the reviews. Please find below a detailed point-by-point reply to the comments and suggestions to reviewer 1.

**Point-by-point response to the reviews:**

- Line 200, (5): I think this formula might be misleading. I cannot find a reference to or a description of Nexcretionrate(a) anywhere else in the text and tables and I think (especially when referring to Table 2) Nexcretionrate(a) should be changed to (1-Nretained). We agree, Nexcretionrate(a) needs to be expressed rather as (1-Nretained). It has been changed in the text.

- Line 341: change sensibility to sensitivity : Change done and modified also in the abstract (line 14).

**Response to the notification from 12 of December 2022:** The tables inserted as figures in the supplementary material have been renamed as tables and the corresponding references in the main text have been corrected.

**Response to the Topical Editor from 14 of January 2023:** The DOI creation is in progress and will be given to the Editor as soon as it is transmitted to the authors. In the meantime, a specific page dedicated to the ORCHIDEE code used has been also created : https://data.ipsl.fr/catalog/srv/eng/catalog.search#/metadata/db1cf5ce-6fd2-4b4c-a3d1-598e2283c19d